# Two Sides of One Coin: the Limits of Untuned SGD and the Power of Adaptive Methods

**Junchi Yang**[*]
Department of Computer Science
ETH Zurich, Switzerland
junchi.yang@inf.ethz.ch

**Xiang Li**[*]
Department of Computer Science
ETH Zurich, Switzerland
xiang.li@inf.ethz.ch

**Ilyas Fatkhullin**
Department of Computer Science
ETH Zurich, Switzerland
ilyas.fatkhullin@ai.ethz.ch

**Niao He**
Department of Computer Science
ETH Zurich, Switzerland
niao.he@inf.ethz.ch

## Abstract

The classical analysis of Stochastic Gradient Descent (SGD) with polynomially decaying stepsize $\eta_t = \eta/\sqrt{t}$ relies on well-tuned $\eta$ depending on problem parameters such as Lipschitz smoothness constant, which is often unknown in practice. In this work, we prove that SGD with arbitrary $\eta > 0$, referred to as *untuned SGD*, still attains an order-optimal convergence rate $\widetilde{\mathcal{O}}(T^{-1/4})$ in terms of gradient norm for minimizing smooth objectives. Unfortunately, it comes at the expense of a catastrophic exponential dependence on the smoothness constant, which we show is unavoidable for this scheme even in the noiseless setting. We then examine three families of adaptive methods — Normalized SGD (NSGD), AMSGrad, and AdaGrad — unveiling their power in preventing such exponential dependency in the absence of information about the smoothness parameter and boundedness of stochastic gradients. Our results provide theoretical justification for the advantage of adaptive methods over untuned SGD in alleviating the issue with large gradients.

## 1 Introduction

In this work, we study the stochastic optimization problem of the form:

$$\min_{x \in \mathbb{R}^d} f(x) = \mathbb{E}_{\xi \sim P} \left[ F(x; \xi) \right],$$

where $P$ is an unknown probability distribution, and $f : \mathbb{R}^d \to \mathbb{R}$ is an $\ell$-Lipschitz smooth function and can be non-convex. In the context of machine learning, $\xi$ typically represent an individual training sample drawn from the data distribution $P$, and $x$ denotes the weights of the model.

Stochastic Gradient Descent (SGD), originating from the seminal work [58], performs the following update iteratively:

$$x_{t+1} = x_t - \eta_t \nabla F(x_t; \xi_t),$$

where $\eta_t > 0$ presents a positive stepsize, and $\nabla F(x_t; \xi_t)$ is an unbiased stochastic gradient. SGD has shown remarkable empirical success in many modern machine learning applications, e.g., [7, 62]. Its efficiency is usually attributed to its cheap per iteration cost and the ability to operate in an

37th Conference on Neural Information Processing Systems (NeurIPS 2023).

---

[*]Equal contribution.

Table 1: Complexities of finding an $\epsilon$-stationary point for SGD, NSGD [53], NSGD-M [14], AMSGrad-norm (norm version of AMSGrad [57]), and AdaGrad-norm [61]. We only assume $f$ is $\ell$-smooth, and unbiased stochastic gradients have bounded variance $\sigma^2$. Hyper-parameters (e.g., $\gamma$ and $\eta$) are untuned. Here, $\widetilde{\mathcal{O}}$ and $\Omega$ hide polynomial terms in problem parameters and hyper-parameters. The bounds are with respect to specific algorithms and stepsizes, and lower bounds for general first-order methods still hold [11, 3]. We list each algorithm's effective stepsize at iteration $t$ directly below its name.

| Algorithms | Upper bound; deterministic | Lower bound; deterministic | Upper bound; stochastic | Lower bound; stochastic |
|---|---|---|---|---|
| SGD (Eq. 1) $\frac{\eta}{\sqrt{t+1}}$ | $\widetilde{\mathcal{O}}\left((4e)^{2(\eta\ell)^2}\epsilon^{-4}\right)$ [Thm. 1, 6] | $\Omega\left((8e)^{\eta^2\ell^2/8}\epsilon^{-4}\right)$ [Thm. 2] | $\widetilde{\mathcal{O}}\left((4e)^{2(\eta\ell)^2}\epsilon^{-4}\right)$ [Thm. 1, 6] | $\Omega\left((8e)^{\eta^2\ell^2/8}\epsilon^{-4}\right)$ [Thm. 2] |
| NSGD (Alg. 3) $\frac{\gamma}{\sqrt{t+1}\|g(x_t;\xi_t)\|}$ | $\widetilde{\mathcal{O}}\left(\epsilon^{-2}\right)$ [14] & [Prop. 1] | $\Omega\left(\epsilon^{-2}\right)$ [11] | N/A due to lower bound | Nonconvergent [Thm. 3] |
| NSGD-M (Alg. 1) $\frac{\gamma}{(t+1)^\alpha\|g_t\|}$ | $\widetilde{\mathcal{O}}\left(\epsilon^{-2}\right), \alpha = 1/2$ [14] & [Prop. 1] | $\Omega\left(\epsilon^{-2}\right)$ [11] | $\widetilde{\mathcal{O}}\left(\epsilon^{-4}\right), \alpha = 3/4$ [14] & [Prop. 1] | $\Omega\left(\epsilon^{-4}\right)$ [3] |
| AMSGrad-norm (Alg. 2) $\frac{\gamma}{\sqrt{(t+1)\hat{v}_{t+1}^2}}$ | $\widetilde{\mathcal{O}}\left(\epsilon^{-4}\right)$ [Thm. 5, 7] | $\Omega\left(\epsilon^{-4}\right)$ [Thm. 8] | N/A due to lower bound | $\Omega\left(\epsilon^{-\frac{2}{1-\zeta}}\right) \forall\zeta \in (\frac{1}{2}, 1)$ [Thm. 4] |
| AdaGrad-norm (Alg. 5) $\frac{\eta}{\sqrt{v_0^2+\sum_{k=0}^t\|g(x_k;\xi_k)\|^2}}$ | $\widetilde{\mathcal{O}}\left(\epsilon^{-2}\right)$ [68] & [Prop. 3] | $\Omega\left(\epsilon^{-2}\right)$ [11] | $\widetilde{\mathcal{O}}\left(\epsilon^{-4}\right)$ [68] & [Prop. 3] | $\Omega\left(\epsilon^{-4}\right)$ [3] |

online fashion, making it suitable for large-scale problems. However, empirical evidence also reveals undesirable behaviors of SGD, often linked to challenges in selecting appropriate stepsizes. In particular, a number of works report the *gradient explosion* effect [6, 55, 24] during the initial phase of training, which may eventually lead to divergence or prohibitively slow convergence. The phenomenon is also observed in our experiments (see Figure 1(f)) when the stepsize is poorly chosen. Unfortunately, this phenomenon has not been well understood from a theoretical point of view. The classical analysis of SGD in the smooth non-convex case [22], prescribes to select a non-increasing sequence of stepsizes $\{\eta_t\}_{t\geq 1}$ with $\eta_1 < 2/\ell$. In particular, the choice $\eta_t = 1/(\ell\sqrt{t})$, guarantees[2] to find a point $x$ with $\mathbb{E}\left[\|\nabla f(x)\|\right] \leq \epsilon$ after $\mathcal{O}\left(\epsilon^{-4}\right)$ stochastic gradient calls, which is also known to be unimprovable in the smooth non-convex setting unless additional assumptions are made [3, 17].

However, the bound on the smoothness parameter $\ell$ is usually not readily available for practitioners, and the limited computing power usually prevents them from exhaustive tuning to find the best stepsize. It is therefore important to provide a theoretical understanding for SGD with an arbitrary stepsize (which we refer to as *untuned SGD*) that is agnostic to the problem parameter. The following intriguing question remains elusive in the stochastic optimization literature:

> *How does untuned SGD with a decaying stepsize $\eta_t = \eta/\sqrt{t}$ perform when $\eta$ is independent of the smoothness parameter? How can we explain the undesirably large gradients encountered in training with SGD?*

Recently, there has been a surge of interest in adaptive gradient methods such as Adam [35], RMSProp [28], AdaDelta [69], AMSGrad [57], AdaGrad [18], Normalized SGD [26], and many others. These methods automatically adjust their stepsizes based on past stochastic gradients, rather than using pre-defined iteration-based schedules. Empirically, they are observed to converge faster than SGD and mitigate the issue of gradient explosion across a range of problems, even without explicit knowledge of problem-specific parameters [35, 46, 55]. Figure 1 provides a basic illustration of performance differences between SGD with $\eta_t = 1/\sqrt{t}$ stepsizes and adaptive schemes such as AdaGrad and Normalized SGD with momentum (NSGD-M) [14]. Notably, when the initial stepsize is too large (relative to $1/\ell$ value), SGD reaches the region with *large gradients*, while adaptive methods do not

---

[2]Given access to unbiased stochastic gradient oracle with bounded variance.

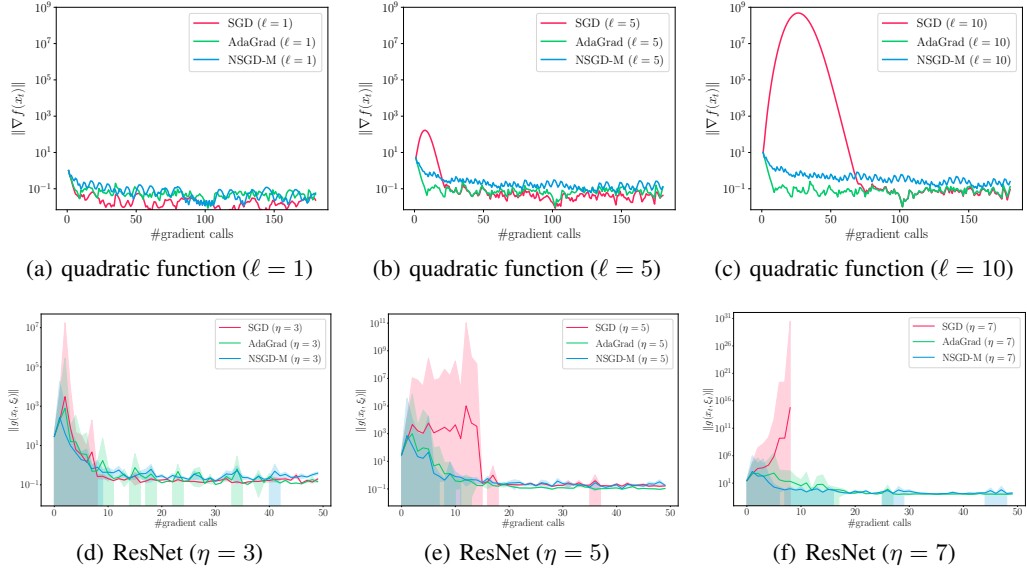

Figure 1: Comparison of SGD, AdaGrad, and NSGD-M on a quadratic function $f(x) = \ell x^2/2$ and a deep neural network. SGD employs a diminishing stepsize of $\eta/\sqrt{t}$, while the stepsizes for AdaGrad and NSGD-M are specified in Propositions 2 and 3, respectively. For experiments on the quadratic function, we set $\eta = 1$ for all methods and test with 3 different values of $\ell$. For the figures in the second row, We train a 50-layer ResNet [27] on the CIFAR-10 dataset [36] using cross-entropy loss. Due to the large randomness observed for this setting, we run each combination of optimizer and stepsize 10 times and plot both the average and variance. Note that the scale of the y-axis varies for each sub-figure and SGD diverges to NaN when $\eta = 7$.

suffer from such effect. However, the theoretical benefits of adaptive methods over SGD remain unclear. A large number of existing analyses of adaptive methods assume bounded gradients, or even stochastic gradients, precluding not only a fair comparison with SGD whose convergence does not need bounded gradient but also the possibility to explain their benefit when facing gradient explosions. While recent developments show that AdaGrad-type methods [20, 68] can attain a $\widetilde{\mathcal{O}}\left(\epsilon^{-4}\right)$ sample complexity under the same standard assumptions as for SGD analysis, there is still a lack a good explanation for the substantial performance gap observed in practice despite SGD with well-tuned stepsizes theoretically achieving the lower complexity bound. We will address the following open question:

> *Can we justify the theoretical benefits of adaptive methods over untuned SGD for smooth non-convex problems without assuming bounded gradients?*

Consequently, this work is grounded on the premise that gradients may be not bounded and that hyper-parameters are independent of the problem parameters. . The main contributions are as follows:

- We show that untuned SGD with diminishing stepsizes $\eta_t = \eta/\sqrt{t}$ finds an $\epsilon$-stationary point of an $\ell$-smooth function within $\widetilde{\mathcal{O}}((\ell^2 + \sigma^4\eta^4\ell^4)(4e)^{2\eta^2\ell^2}\epsilon^{-4})$ iterations for any $\eta > 0$. Here, $\sigma^2$ corresponds to the variance of the stochastic gradient. Although this classical algorithm converges and has the optimal dependence on $\epsilon$, we further show that the disastrous exponential term in $\eta^2\ell^2$ is unavoidable even when the algorithm has access to exact gradients. This explains its proneness to gradient explosion when the problem parameter is unknown. Previous analyses have failed to capture this exponential term, since they assume $\eta$ is well-tuned to be $\Theta(1/\ell)$.

- AMSGrad, which was proposed to fix the nonconvergence of Adam, is not yet fully understood, with previous analyses depending on *bounded stochastic gradients*. We show that AMSGrad (norm version) is free from exponential constants in the deterministic setting without tuning, in stark contrast with SGD. Surprisingly, in the stochastic setting when the stochastic gradients are

unbounded, we show that AMSGrad may converge at an arbitrarily slow polynomial rate. To the best of our knowledge, these are the first results for AMSGrad without assuming bounded gradients.

- To further illuminate the advantages of adaptive methods, we re-examine the results for Normalized Gradient Descent (NGD), Normalized SGD with momentum (NSGD-M) from [14] and AdaGrad-norm from [68], considering stepsize independent of the problem parameters similar to untuned SGD. They all achieve near-optimal complexities while shredding off the exponential factor. As a side result, we provide a strong non-convergence result of NSGD without momentum under any bounded stepsizes, which might be of independent interest.

Our findings contribute a fresh understanding of the performance gap between SGD and adaptive methods. Albeit with a near-optimal rate, untuned SGD is vulnerable to gradient explosion and slow convergence due to a large exponential constant in its complexity, which can be circumvented by several adaptive methods. To the best of our knowledge, this substantial difference is unaddressed in the previous literature, because the majority of analyses for SGD and adaptive methods turn to either well-tuned stepsize based on problem parameters or the assumption of bounded gradients. Part of our results are summarized in Table 1, and full results for a broader range of stepsizes can be found in Table 2 in the appendix.

## 1.1 Related Work

**SGD in nonconvex optimization.** Stochastic approximation methods and SGD in particular have a long history of development [58, 34, 8, 13, 51, 56]. When the objective is $\ell$-smooth and the gradient noise has bounded variance $\sigma^2$, Ghadimi and Lan [22] and Bottou et al. [9] prove that if stepsize $\eta_t = \eta/\sqrt{T}$, where $\eta = \eta(\ell, \sigma^2)$ and $T$ is the total iteration budget, then SGD can find an $\epsilon$-stationary point within $\mathcal{O}(\ell\sigma^2\epsilon^{-4})$ iterations. Similar complexity (up to a logarithmic term) can also be achieved by decaying stepsizes $\eta/\sqrt{t}$ [22, 17, 65]. This result was later shown to be optimal for first-order methods under these assumptions [3]. Several works consider various relaxations of the stochastic oracle model with bounded variance, for instance, biased oracle [2] and relaxed growth condition [9]. However, these results also heavily rely on sufficiently small $\eta$, e.g., $\eta \leq 1/\ell$, and the convergence behavior in the large $\eta$ regime is rarely discussed. Remarkably, Lei et al. [39] characterize the convergence of SGD under individual smoothness and unbiased function values. They consider Robbins-Monro stepsize schemes, which includes $\eta/t^\alpha$ when $\alpha > 1/2$, and derive $\mathcal{O}(\epsilon^{\frac{2}{\alpha-1}})$ sample complexity including an exponential dependence on the smoothness parameter. Unlike [39], we focus on the standard assumptions and derive better dependency in smoothness constant when $\alpha > 1/2$. Importantly, we further justify that the exponential constants are unavoidable with a lower bound.

**Adaptive methods.** We focus on methods directly using gradients to adjust stepsize, rather than other strategies like backtracking line search [4]. Normalized Gradient Descent (NGD) was introduced by [53] for quasi-convex functions. Hazan et al. [26] apply NGD and NSGD with minibatch to the class of locally-quasi-convex functions. Later, Cutkosky and Mehta [14] and Zhao et al. [74] prove NSGD with momentum or minibatch can find an $\epsilon$-stationary point in smooth nonconvex optimization with sample complexity $\mathcal{O}(\epsilon^{-4})$. AdaGrad was introduced in the online convex optimization [18, 48]. In nonconvex optimization, AdaGrad and its scalar version, AdaGrad-norm [61], achieve competitive convergence rates with SGD [66, 43, 31, 44]. RMSProp [28] and Adam [35] use the moving average of past gradients, but may suffer from divergence without hyper-parameter tuning [57]. Recently, it was shown in the finite-sum setting that they converge to a neighborhood, whose size shrinks to 0 by tuning hyper-parameters [60, 72]. However, most of the results on AdaGrad and Adam-type algorithms assume both Lipschitz and bounded gradients [75, 12, 16, 66, 76]. Very recently, Faw et al. [20] and Yang et al. [68] independently show that AdaGrad-norm converges without assuming bounded gradients and without the need for tuning, attaining a sample complexity of $\widetilde{\mathcal{O}}(\epsilon^{-4})$.

**SGD v.s. adaptive methods.** Despite similar complexities, adaptive methods typically converge faster than SGD in practice [10, 47] and are widely used to prevent large gradients [55, 23]. Various attempts have been made to theoretically explain these differences. Some suggest that the advantage of adaptive algorithms is their ability to achieve order-optimal rates without knowledge of problem parameters such as smoothness and noise variance [66, 40, 30]. Other studies investigate the faster

escape from saddle points by adaptive methods [41, 50, 67]. The importance of coordinate-wise normalization in Adam has also been highlighted [5, 37]. Furthermore, the influence of heavy-tail noise on the performance of adaptive methods is studied [71]. However, most previous works do not provide an explanation for the faster convergence of adaptive methods in terms of sample complexity. Notably, Zhang et al. [70] and Wang et al. [64] explain the benefits of gradient clipping and Adam under a relaxed smoothness assumption, a setting where SGD with non-adaptive stepsizes may not converge. In contrast, we analyze SGD and several adaptive methods under standard smoothness and noise assumptions, distinguishing it from the recent work of Wang et al. [64] that focuses on one variant of Adam for finite-sum problems with individual relaxed smoothness and random shuffling.

## 2  Problem Setting

Throughout this work, we focus on minimizing an $\ell$-smooth function $f : \mathbb{R}^d \to \mathbb{R}$. We have access to a stochastic gradient oracle that returns $g(x; \xi)$ at any point $x$, and we make the following standard assumptions in nonconvex optimization.

**Assumption 1** (smoothness). *Function $f(x)$ is $\ell$-smooth with $\ell > 0$, that is, $\|\nabla f(x_1) - \nabla f(x_2)\| \le \ell \|x_1 - x_2\|$ for any $x_1$ and $x_2 \in \mathbb{R}^d$.*

**Assumption 2** (stochastic gradients). *The stochastic gradient $g(x; \xi)$ is unbiased and has a bounded variance, that is, $\mathbb{E}_\xi [g(x; \xi)] = \nabla f(x)$ and $\mathbb{E}_\xi \left[ \|g(x; \xi) - \nabla f(x)\|^2 \right] \le \sigma^2$ for any $x \in \mathbb{R}^d$.*

We present the general scheme of SGD with initial point $x_0$ and a stepsize sequence $\{\eta_t\}_{t=0}^\infty$:

$$x_{t+1} = x_t - \eta_t \, g(x_t; \xi_t). \tag{1}$$

Some commonly used stepsizes include polynomially and geometrically decaying stepsize, constant stepsize, cosine stepsize, etc. When the stepsize depends on the instantaneous or past gradients, i.e., $\{g(x; \xi_k)\}_{k \le t}$, we call it adaptive stepsize, namely Normalized SGD [26], AdaGrad [18], Adam [35], AMSGrad [57], etc. In some adaptive methods, momentum is also considered, replacing $g(x_t; \xi_t)$ in (1) with a moving average $m_{t+1}$ of the past stochastic gradients (see Section 4 for more details). To set the stage for our analysis, we assume that $f(x_0) - \min_{x \in \mathbb{R}^d} f(x) \le \Delta$, where $\Delta$ represents the initial gap. Given that the function class of interest is nonconvex, we aim to find an $\epsilon$-stationary point $x$ with $\mathbb{E}[\|\nabla f(x)\|] \le \epsilon$.

## 3  Convergence of Untuned SGD

In this section, we focus on SGD with the decaying stepsize:

$$\eta_t = \frac{\eta}{\sqrt{t+1}},$$

where $\eta > 0$ is the initial stepsize. Most convergent analysis requires $\eta < 2/\ell$ [22, 9] so that there is "sufficient decrease" in function value after each update, and if $\eta$ is carefully chosen, it can achieve the near-optimal complexity of $\widetilde{\mathcal{O}}(\ell \epsilon^{-4} \sigma^2)$ [3]. Nevertheless, as the smoothness parameter is usually unknown, providing guarantees with optimal $\eta$ or assuming $\eta$ to be problem-dependent does not give enough insights into practical training with SGD. Hence we are interested in its convergence behavior in both small and large initial stepsize regimes, i.e., $\eta \le 1/\ell$ and $\eta > 1/\ell$.

**Theorem 1.** *Under Assumptions 1 and 2, if we run SGD with stepsize $\eta_t = \frac{\eta}{\sqrt{t+1}}$, where $\eta > 0$,*

$$\frac{1}{T} \sum_{t=0}^{T-1} \mathbb{E}\|\nabla f(x_t)\|^2 \le \begin{cases} 2A\eta^{-1} T^{-\frac{1}{2}}, & \text{when } \eta \le 1/\ell, \\ 4\sqrt{2}\ell A (4e)^\tau (\pi T)^{-\frac{1}{2}}, & \text{when } \eta > 1/\ell, \end{cases}$$

*where $\tau = \lceil \eta^2 \ell^2 - 1 \rceil$ and $A = \left( \Delta + \frac{\ell \sigma^2 \eta^2}{2}(1 + \log T) \right)$.*

This theorem implies that when the initial stepsize $\eta > 1/\ell$, SGD still converges with a sample complexity of $\widetilde{\mathcal{O}}((\ell^2 + \sigma^4 \eta^4 \ell^4)(4e)^{2\eta^2 \ell^2} \epsilon^{-4})$. Although the dependency in the target accuracy $\epsilon$ is

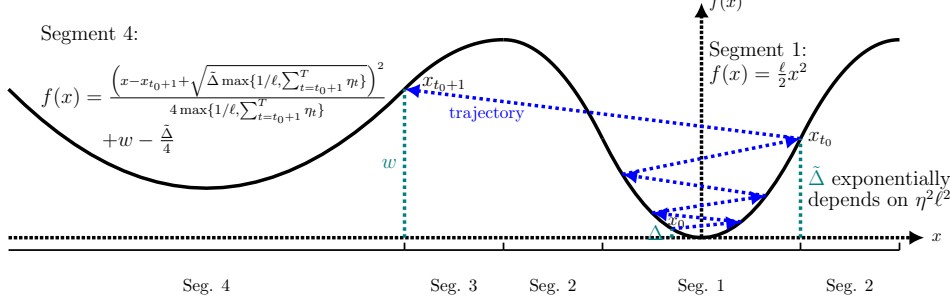

Figure 2: Demonstration of the constructed function used to prove the lower bound. The definitions of the segments comprising the function can be found in Appendix B.2.

near-optimal, it includes a disastrous exponential term in $\eta^2\ell^2$. This is due to polynomially decaying stepsizes: in the first stage before $\tau = \lceil(\eta\ell)^2 - 1\rceil$ iterations, the function value and gradients may keep increasing in expectation until reaching an exponential term in $\eta^2\ell^2$, which is in stark contrast with adaptive methods that we will see in Section 4; in the second stage after $t \geq \tau$, the stepsize is small enough to decrease the function value in expectation at a rate of $1/\sqrt{T}$ up to a small term in $\sigma$.

If we arbitrarily select $\eta = \Theta(1)$, untuned SGD may induce large gradients growing exponentially in $\ell$ in the first stage, as observed in Figure 1. Moreover, deriving the dependence in hyper-parameter $\eta$ is crucial for assessing the effort required in its tuning: SGD with an $\eta$ that is $c > 1$ times larger than the optimally tuned value can yield a gradient norm that is $\exp(\mathrm{poly}(c))$ times larger in the convergence guarantee.

To the best of our knowledge, there is limited study for non-asymptotic analysis of untuned SGD under the same assumptions. Moulines and Bach [49] study untuned SGD under individual smoothness and convexity assumptions, i.e., $g(x;\xi)$ is Lipschitz continuous and $F(x;\xi)$ is convex almost surely. They showed a convergence rate of $\mathcal{O}(1/T^{1/3})$ for the last iterate and a rate arbitrarily close to $\mathcal{O}(1/T^{1/2})$ for the averaged iterate. Later, Fontaine et al. [21] provide $\widetilde{\mathcal{O}}(1/T^{1/2})$ convergence rate for untuned SGD in the convex setting, albeit without an explicit dependency in $\ell$ and $\eta$. In the nonconvex setting, Khaled and Richtárik [33] showed, using SGD with constant stepsize $\eta > 0$, the squared gradient $\min_{t<T} \mathbb{E}\|\nabla f(x_t)\|^2$ converges at a rate of $\mathcal{O}\left(\ell\sigma\eta + (1 + \ell^2\eta^2)^T \Delta/(\eta T)\right)$. When the total number of iterations $T$ is predetermined, selecting a constant stepsize $\eta = \eta_0/\sqrt{T}$ results in the rate $\mathcal{O}\left(\frac{\ell\sigma\eta_0}{\sqrt{T}} + \frac{(1+\ell^2\eta_0^2/T)^T \Delta}{\eta_0\sqrt{T}}\right) \leq \mathcal{O}\left(\frac{\ell\sigma\eta_0}{\sqrt{T}} + \frac{e^{\ell^2\eta_0^2}\Delta}{\eta_0\sqrt{T}}\right)$, which exhibits the same dependence on $T$ as Theorem 1 and also includes an exponential constant. However, our focus is on settings where $T$ is not known in advance.

**Remark 1.** *We consider stepsize in the order of $1/\sqrt{t}$, because it is known for SGD to achieve the best dependency in $\epsilon$ for nonconvex optimization [17] and easier to compare with adaptive stepsizes. We also present the convergence results for more general polynomially decaying stepsizes, i.e., $\eta_t = \frac{\eta}{(t+1)^\alpha}$ with $0 < \alpha < 1$, in Theorem 6 of the appendix. There exists a trade-off between convergence speed $\mathcal{O}(1/T^{\frac{1-\alpha}{2}})$ and the exponential term in $(\eta\ell)^{1/\alpha}$ for $\alpha \in [1/2, 1)$. Intuitively, larger $\alpha$ leads to a shorter time in adapting to $1/\ell$ stepsize but a slower convergence rate. We do not consider constant stepsize, i.e., $\alpha = 0$, because it is well known to diverge even in the deterministic setting if the stepsize is agnostic to the problem parameter [52, 1].*

The question arises as to whether the exponential term is necessary. In the following, we provide a lower bound for SGD under this choice of stepsize.

**Theorem 2.** *Fixing $T \geq 1, \eta > 0, \ell > 0$ and $\Delta > 0$ that $\eta\ell \geq 5$, there exists a $\ell$-smooth function $f : \mathbb{R} \to \mathbb{R}$ and an initial point $x_0$ with $f(x_0) - f^* \leq \Delta$ such that if we run Gradient Descent with stepsize $\eta_t = \frac{\eta}{\sqrt{t+1}}$, then for $t \leq t_0 = \lfloor \eta^2\ell^2/16 - 1 \rfloor$,*

$$|\nabla f(x_t)| \geq \sqrt{\frac{2\ell\Delta}{3\sqrt{t}}}(8e)^{t/2} \text{ and } |\nabla f(x_{t_0})| \geq \sqrt{\frac{8\Delta}{3\eta}}(8e)^{\eta^2\ell^2/32-4};$$

*if $T > t_0$, then for $t_0 < t \leq T$,*

$$|\nabla f(x_t)| \geq \frac{1}{4}\sqrt{\tilde{\Delta}} \min\left\{\ell^{1/2}, (2\eta)^{-1/2}T^{-1/4}\right\}, \ where \ \tilde{\Delta} \geq \frac{4}{3\eta\ell}(8e)^{\eta^2\ell^2/16-2}\Delta.$$

This theorem suggests that Gradient Descent (GD) with decaying stepsize $\eta/\sqrt{t+1}$ needs at least $\Omega(\eta^{-4}\ell^{-2}(8e)^{\eta^2\ell^2/8}\epsilon^{-4})$ iterations to find an $\epsilon$-stationary point in the large initial stepsize regime. Therefore, it validates that an exponential term in $\eta^2\ell^2$ *multiplied by* $\epsilon^{-4}$ is not avoidable even in the deterministic setting. It is crucial to note that our result is limited to untuned (S)GD with this particular stepsize scheme.

It is worth pointing out that the existing lower bounds for first-order methods [3] and SGD [17] do not contain any exponential terms. On the other hand, Vaswani et al. [63] established a lower bound in the strongly convex setting for GD with exponentially decreasing stepsizes $(1/T)^{t/T}\eta$, where $T$ denotes the total number of iterations and $\eta > 0$. With a quadratic function, they showed that if $\eta > 3/\ell$, the distance to the optimal solution $\|x_t - x^*\|$ exhibits exponential growth in $t$ during the initial $\widetilde{\Theta}(T/\log T)$ iterations. However, this exponential growth is rapidly mitigated afterward, resulting in no exponential term at the final iterate $T$. This behavior starkly contrasts with our nonconvex setting.

We illustrate our hard instance for Theorem 2 in Figure 2, which is one-dimensional. The algorithm starts from a valley of the function $f(x) = \ell x^2/2$, i.e., Segment 1. Because of the large initial stepsize and steep slope, in the first $t_0$ iterations, Gradient Descent increases the function value as large as $\tilde{\Delta} = \Omega\left((8e)^{\eta^2\ell^2/16}\Delta\right)$. Then the iterate $x_{t_0+1}$ jumps to the top of a very flat valley, i.e., Segment 4, so that Gradient Descent decreases the gradient as slowly as $\Omega(T^{-1/4})$.

*Why do not we assume gradients to be bounded?* The assumption on bounded gradients is not satisfied even for the simple function $f(x) = \ell x^2/2$. When training neural networks, gradient explosion is often observed [55, 59], which directly suggests that this assumption is not satisfied or only satisfied with a numerically large constant. In Proposition 4 in the appendix, we also provide a simple proof for the convergence under the additional assumption of bounded gradient, i.e., $\|\nabla f(x)\| \leq G$ for all $x$, attaining a sample complexity of $\widetilde{\mathcal{O}}(\eta^2\ell^2 G^4\sigma^2\epsilon^{-4})$ without any information about problem parameters. However, compared with Theorem 1 and 2, constant $G$ hides the exponential term. In Figure 1, we observe that the gradient bound along the trajectory of non-adaptive stepsize can be much larger than that of adaptive stepsize even if starting from the same initial point, so assuming bounded gradient will obscure the difference between them.

## 4 Power of Adaptive Methods

In this section, we focus on the convergence behaviors of adaptive methods, which adjust their stepsizes based on the observed gradients. In particular, when arriving at a point with a large gradient, adaptive methods automatically decrease their stepsizes to counter the effect of possible gradient increase; to list a few, Normalized SGD [26], AdaGrad [18], Adam [35]. Since the analysis for adaptive methods is usually on a case-by-case basis, we will examine three examples – Normalized SGD, AMSGrad-norm, and AdaGrad-norm – to establish a universal observation that they avoid exponential dependency in $\ell$ without tuning. Although many existing analyses rely on bounded gradients (and function values) or information on problem parameters, we will abandon such assumptions as noted in the previous section. We focus on the norm instead of the coordinate-wise version of adaptive methods, which means each coordinate adopts the same stepsize, because the norm version is usually dimension-independent in the complexity, and is also widely used in both theory and practice [73, 45, 43, 38, 54, 32].

### 4.1 Family of Normalized SGD

Normalized (Stochastic) Gradient Descent [53, 26], referred to as NGD and NSGD, is one of the simplest adaptive methods. It takes the stepsize in (1) to be normalized by the norm of the current (stochastic) gradient:

$$\eta_t = \frac{\gamma_t}{\|g(x_t; \xi_t)\|},$$

where $\{\gamma_t\}_{t\geq 0}$ is a sequence of positive learning rate. Cutkosky and Mehta [14] and Zhao et al. [74] show that NSGD with $\gamma_t = \gamma/\sqrt{T}$ can find an $\mathcal{O}(1/\sqrt{T} + \sigma)$-stationary point. In order to compare fairly with untuned SGD with decaying stepsize, we present a modification with decaying $\gamma_t = \gamma/\sqrt{t+1}$ in NSGD.

**Proposition 1.** *Under Assumption 1 and 2, if we run NSGD with $\gamma_t = \frac{\gamma}{\sqrt{t+1}}$, then for any $\gamma > 0$,*

$$\frac{1}{T}\sum_{t=0}^{T-1}\mathbb{E}\|\nabla f(x_t)\| \leq 3\left(\frac{\Delta}{\gamma} + \ell\gamma\log(T)\right)T^{-1/2} + 24\sigma.$$

**NGD.** In the deterministic setting, by Proposition 1, NGD converges to an $\epsilon$-stationary point with a complexity of $\widetilde{\mathcal{O}}((\gamma^{-2} + \gamma^2\ell^2)\epsilon^{-2})$ for any $\gamma > 0$, which importantly does not include any exponential term. Thus, even if the initial stepsize is not small enough, it does not result in a catastrophic gradient explosion.

**NSGD.** In the stochastic setting, Proposition 1 implies that NSGD can find an $\epsilon$-stationary point only when the noise variance is small enough, i.e., $\sigma \leq \mathcal{O}(\epsilon)$. This is not the consequence of a loose analysis. Hazan et al. [26] show that NSGD with constant $\gamma_t \equiv \gamma$ does not converge when the mini-batch size is smaller than $\Theta(\epsilon^{-1})$ for a non-smooth convex function. Here we provide a non-convergence result in the gradient norm with a smooth objective for all uniformly bounded stepsizes. The intuition behind this is illustrated in Figure 3 in the appendix, where $\mathbb{E}_\xi\, g(x; \xi)/\|g(x; \xi)\|$ can easily vanish or be in the opposite direction of $\nabla f(x)$ under certain noises.

**Theorem 3.** *Fixing $\ell > 0$, $\sigma > 0$, $\epsilon > 0$, $\Delta > 0$ and stepsize sequence $\{\gamma_t\}_{t=0}^{\infty}$ with $\gamma_t \leq \gamma_{\max}$ that $\epsilon^2 < \min\{\sigma^2, 2\ell\Delta, 2\Delta(\sigma - \epsilon)/\gamma_{\max}\}$, there exists an $\ell$-smooth convex function $f$, initial point $x_0$ with $f(x_0) - \min_x f(x) \leq \Delta$ and zero-mean noises with $\sigma^2$ variance such that the output from NSGD satisfies $\mathbb{E}\|\nabla f(x_t)\| \geq \epsilon$ for all $t$.*

This theorem implies that, given a fixed function class $(\ell, \Delta, \sigma)$ and any sequence $\{\gamma_t\}_t$ uniformly upper bounded by $\gamma_{\max}$, NSGD cannot converge to an arbitrarily small $\epsilon$. Specifically, the expected gradient norm will always stay larger than $\min\{\sigma, \sqrt{2\ell\Delta}, \gamma_{\max}^{-1}(-\Delta + \sqrt{\Delta^2 + 2\Delta\gamma_{\max}\sigma})\}$. Most $\{\gamma_t\}_t$ used in practice is upper bounded, e.g., constant or decreasing sequences. The condition $\epsilon^2 < 2\ell\Delta$ is necessary by noting that $\|\nabla f(x_0)\|^2 \leq 2\ell[f(x_0) - \min_x f(x)] \leq 2\ell\Delta$. Considering $\gamma_t = 1/\sqrt{t+1}$, when $\Delta \geq \sigma$ and $\sqrt{2\ell\Delta} \geq \sigma$, it matches with Proposition 1, where NSGD can only converge to a $\Theta(\sigma)$-stationary point. Since Sign-SGD and NSGD coincide for one-dimensional objectives, our non-convergent example also applies to Sign-SGD. This sheds light on why increasing batch size improves Normalized and Sign-SGD [74, 37]. However, these methods are generally different in higher dimensions, as Karimireddy et al. [29] show that sign-SGD may not converge even with a full batch.

**NSGD with momentum.** While NSGD may not always converge, Cutkosky and Mehta [14] introduced NSGD with momentum (NSGD-M) presented in Algorithm 1 with constant $\gamma_t \equiv \gamma$. We provide the following modification with diminishing $\gamma_t$ that eliminates the need to specify the total number of runs beforehand.

**Proposition 2.** *Under Assumptions 1 and 2, if we run NSGD-M with $\alpha_t = \frac{\sqrt{2}}{\sqrt{t+2}}$ and $\gamma_t = \frac{\gamma}{(t+1)^{3/4}}$, then for any $\gamma > 0$,*

$$\frac{1}{T}\sum_{t=0}^{T-1}\mathbb{E}\|\nabla f(x_t)\| \leq C\left(\frac{\Delta}{\gamma} + (\sigma + \ell\gamma)\log(T)\right)T^{-\frac{1}{4}},$$

*where $C > 0$ is a numerical constant.*

It implies that NSGD-M attains a complexity of $\widetilde{\mathcal{O}}((\gamma^{-4} + \gamma^4\ell^4)\epsilon^{-4})$ for any $\gamma > 0$. Compared with Theorem 1 and 2, NSGD-M not only achieves near-optimal dependency in the target accuracy $\epsilon$, but also shreds the exponential term when the hyper-parameter is agnostic to smoothness constant.

| **Algorithm 1** NSGD-M | **Algorithm 2** AMSGrad-norm |
|---|---|
| 1: **Input:** initial point $x_0$, stepsize sequence $\{\gamma_t\}$, momentum sequence $\{\alpha_t\}$, and initial momentum $g_0$. | 1: **Input:** initial point $x_0$, momentum parameters $0 \leq \beta_1 < 1$ and $0 \leq \beta_2 \leq 1$, stepsize sequence $\{\gamma_t\}$ and initial momentum $m_0$ and $v_0 > 0$. |
| 2: **for** $t = 0, 1, 2, ...$ **do** | 2: $\hat{v}_0 = v_0$ |
| | 3: **for** $t = 0, 1, 2, ...$ **do** |
| 3: $\quad x_{t+1} = x_t - \frac{\gamma_t}{\|g_t\|}g_t$ | 4: $\quad$ sample $\xi_t$ |
| | 5: $\quad m_{t+1} = \beta_1 m_t + (1 - \beta_1)g(x_t; \xi_t)$ |
| 4: $\quad$ sample $\xi_{t+1}$ | 6: $\quad v_{t+1}^2 = \beta_2 v_t^2 + (1 - \beta_2)\|g(x_t; \xi_t)\|^2$ |
| | 7: $\quad \hat{v}_{t+1}^2 = \max\{\hat{v}_t^2, v_{t+1}^2\}$ |
| 5: $\quad g_{t+1} = (1 - \alpha_t)g_t + \alpha_t g(x_{t+1}; \xi_{t+1})$ | 8: $\quad x_{t+1} = x_t - \frac{\gamma_t}{\sqrt{\hat{v}_{t+1}^2}}m_{t+1}$ |
| 6: **end for** | 9: **end for** |

## 4.2 AMSGrad-norm

AMSGrad was introduced by Reddi et al. [57] to fix the possible non-convergence issue of Adam. Notably, current analyses of AMSGrad in the stochastic setting show a convergence rate of $\widetilde{\mathcal{O}}(1/T^{1/4})$, but they rely on the assumption of *bounded stochastic gradients* [12, 75], which is much stronger than assumptions used for SGD analysis. Here, we examine the simpler norm version of AMSGrad, presented in Algorithm 2. We prove that without assuming bounded stochastic gradients, AMSGrad-norm with default $\gamma_t = \gamma/\sqrt{t+1}$ may converge at an arbitrarily slow polynomial rate. In fact, this holds even if the true gradients are bounded. We believe this result is of independent interest.

**Theorem 4.** *For any $\ell > 0$, $\Delta > 0$, $\sigma > 0$ and $T > 1$, there exists a $\ell$-smooth function $f : \mathbb{R}^2 \to \mathbb{R}^2$, $x_0$ with $f(x_0) - \inf_x f(x) \leq \Delta$ and noise distribution $P$ with variance upper bounded by $\sigma^2$, such that if we run AMSGrad-norm with $0 \leq \beta_1 \leq 1$, $0 \leq \beta_2 < 1$ and $\gamma_t = \frac{\gamma}{\sqrt{t+1}}$, we have with probability $\frac{1}{2}$, it holds that*

$$\min_{t \in \{0, 1, ..., T-1\}} \|\nabla f(x_t)\| \geq \sqrt{\frac{\Delta}{16 \max\left\{1/\ell, \frac{\gamma\sqrt{2\Gamma\left(1 - \frac{\zeta}{2}\right)}}{\sigma\left(e\left(\frac{1}{\zeta} - 1\right)\right)^{\frac{\zeta}{2}}(1 - \zeta)\sqrt{1 - \beta_2}}\left(T^{1-\zeta} - \zeta\right)\right\}}}$$

*for any $\frac{1}{2} < \zeta < 1$, where $\Gamma(\cdot)$ denotes the Gamma function.*

The intuition behind this theroem is that since AMSGrad utilizes the maximum norm of past stochastic gradients with momentum in the denominator of stepsizes, some noise distributions enable this maximum norm to increase polynomially, making the stepsizes too small. However, we can still explore its benefit in the deterministic setting. Whether it converges without assuming bounded gradients, to the best of our knowledge, is unknown. Here, for simplicity, we consider AMSGrad-norm without momentum, i.e., $\beta_1 = \beta_2 = 0$.

**Theorem 5.** *Under Assumption 1, if we run AMSGrad-norm with $\gamma_t = \frac{\gamma}{\sqrt{t+1}}$, $v_0 > 0$ and $\beta_1 = \beta_2 = 0$ in the deterministic setting, then for any $\gamma > 0$ and $0 < \alpha < 1$,*

$$\frac{1}{T}\sum_{t=0}^{T-1} \|\nabla f(x_t)\| \leq \begin{cases} T^{-\frac{1}{4}}\sqrt{2\Delta \max\{v_0, \sqrt{2\ell\Delta}\}}\gamma^{-1}, & \text{when } v_0 < \gamma\ell, \\ T^{-\frac{1}{2}}\gamma^2\ell^2 v_0^{-2} + T^{-\frac{1}{4}}\sqrt{2\gamma(M + \Delta)\max\{\gamma\ell, \sqrt{2\ell(M + \Delta)}\}}, & \text{when } v_0 \geq \gamma\ell, \end{cases}$$

*where $M = \ell\gamma^2\left(1 + \log\left(\frac{\ell\gamma}{v_0}\right)\right)$.*

The theorem implies that AMSGrad-norm achieves a complexity of $\widetilde{\mathcal{O}}((\ell^4\gamma^4 + \ell^2 + \ell^3\gamma^2 + \ell\gamma^{-2})\epsilon^{-4})$ with the default $\gamma_t = \Theta(t^{-1/2})$ [57, 12, 25]. Compared with untuned Gradient Descent, it gets rid of the exponential dependency. In the proof, we show that before the first iteration $\tau$ when stepsize $\eta_t$

reduces to $1/\ell$, the accumulated gradient norms $\sum_{t=0}^{\tau-1} \|\nabla f(x_t)\|^2$ are upper bounded polynomially, which is in striking contrast with SGD in Theorem 2. We further provide theoretical guarantees for more general schemes $\frac{\gamma}{(t+1)^\alpha}$ with $0 < \alpha < 1$ in Theorem 7 in the appendix. We also derive matching lower bounds in Theorem 8 for any $0 < \alpha < 1$, and justify that AMSGrad may fail to converge with constant $\gamma_t \equiv \gamma$ (i.e., $\alpha = 0$) if the problem parameter is unknown.

### 4.3 AdaGrad-norm

AdaGrad chooses its stepsize to be inversely proportional to the element-wise accumulated past gradients [18, 48]. Its norm-version, AdaGrad-norm [61, 66], picks stepsize in (1) to be

$$\eta_t = \frac{\eta}{\sqrt{v_0^2 + \sum_{k=0}^{t} \|g(x_k;\xi_k)\|^2}},$$

where $v_0 > 0$. Very recently, AdaGrad is proven to converge in nonconvex optimization without the assumption on bounded gradients or tuning $\eta$ [20, 68]. Although the result in [68] are presented for minimax optimization problems, a similar result follows immediately for minimization problems. We present the following result for the completeness of the paper and to further illustrate the benefits of adaptive methods over SGD.

**Proposition 3.** *Under Assumptions 1 and 2, if we run AdaGrad-norm, then for any $\eta > 0$ and $v_0 > 0$,*

$$\frac{1}{T} \sum_{t=0}^{T-1} \mathbb{E}\|\nabla f(x_t)\| \leq \frac{2A}{\sqrt{T}} + \frac{\sqrt{v_0 A}}{\sqrt{T}} + \frac{2\sqrt{A\sigma}}{T^{\frac{1}{4}}},$$

*where $A = \widetilde{\mathcal{O}}\left(\frac{\Delta}{\eta} + \sigma + \ell\eta\right)$.*

The above result implies a complexity of $\widetilde{\mathcal{O}}\left((\eta^{-2} + \sigma^2 + \eta^2\ell^2)\left(\epsilon^{-2} + \sigma^2\epsilon^{-4}\right)\right)$. Notably, if $\eta$ can be chosen to be $1/\sqrt{\ell}$, it achieves the optimal complexity in both $\ell$ and $\epsilon$ up to logarithmic terms like well-tuned SGD [3]. Even if $\eta$ is agnostic to $\ell$, AdaGrad-norm does not suffer from the exponential term present in untuned SGD. One of the intuitions in the deterministic setting, similar to the AMSGrad-norm, is that the accumulated squared gradient norm before the first iteration with stepsize smaller than $1/\ell$ will be upper bounded by a polynomial term (see Theorem 3.2 in [42]). Another benefit of AdaGrad over other methods is to achieve optimal convergence rates simultaneously in deterministic and stochastic settings with the same hyper-parameters. This is sometimes referred to as "noise adaptivity", which is out of the scope of this paper.

## 5 Conclusion and Future Directions

We convey a crucial message: SGD with a polynomially decaying stepsize can converge at the order-optimal rate, while remaining agnostic to the problem-specific parameter - a notion that may challenge common belief. However, it is subject to an exponential term, which can be avoided by utilizing an appropriate adaptive scheme. We further reveal the similarity and differences between several adaptive schemes; a detailed exposition of these results can be found in Appendix A. This work opens up several avenues for future research. Firstly, it would be valuable to investigate whether the benefits of adaptive methods extend to high-probability convergence guarantees and apply to a broader range of adaptive optimizers. Secondly, removing the assumption of bounded gradients is crucial for a comprehensive analysis of adaptive algorithms, as it can reveal the true dependence on the smoothness parameter $\ell$ and highlight their advantage over SGD. Lastly, examining the impact of adaptive algorithms on optimizing non-smooth nonconvex objectives, which are prevalent in training modern machine learning models, presents an interesting research direction.

## Acknowledgement

The work is supported by ETH research grant and Swiss National Science Foundation (SNSF) Project Funding No. 200021-207343. Ilyas Fatkhullin is partly funded by ETH AI Center.

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

# A Results Summary

Table 2: Comparisons of complexities to find an $\epsilon$-stationary point, i.e., $\mathbb{E}\|\nabla f(x)\| \le \epsilon$, between SGD, NSGD, NSGD-M, AMSGrad-norm and AdaGrad-norm. We only assume $f$ is $\ell$-smooth, and unbiased stochastic gradients have bounded variance $\sigma^2$. Hyper-parameters (e.g., $\gamma$ and $\eta$) are arbitrary and untuned. In this table, $\widetilde{\mathcal{O}}$ and $\Omega$ hide polynomial terms in problem parameters and hyper-parameters, and $\widetilde{\mathcal{O}}$ also hides all logarithmic terms. We use $\eta_t$ to denote the effective stepsize at iteration $t$.

| Algorithms | Upper bound; deterministic | Lower bound; deterministic | Upper bound; stochastic | Lower bound; stochastic |
|---|---|---|---|---|
| SGD (Eq. 1) $\eta_t = \frac{\eta}{(t+1)^\alpha}$ | $\widetilde{\mathcal{O}}\left((4e^{2\alpha})^{\frac{(\eta\ell)^{1/\alpha}}{(1-\alpha)\wedge\alpha}}\epsilon^{\frac{-2}{(1-\alpha)\wedge\alpha}}\right)$ $\alpha \in (0,1)$ [Thm. 1, 6] | $\Omega\left((8e)^{\eta^2\ell^2/8}\epsilon^{-4}\right)$ $\alpha = 1/2$ [Thm. 2] | $\widetilde{\mathcal{O}}\left((4e^{2\alpha})^{\frac{(\eta\ell)^{1/\alpha}}{(1-\alpha)\wedge\alpha}}\epsilon^{\frac{-2}{(1-\alpha)\wedge\alpha}}\right)$ $\alpha \in (0,1)$ [Thm. 1, 6] | $\Omega\left((8e)^{\eta^2\ell^2/8}\epsilon^{-4}\right)$ $\alpha = 1/2$ [Thm. 2] |
| NSGD (Alg. 3) $\eta_t = \frac{\gamma_t}{\|g(x_t;\xi_t)\|}$ | $\widetilde{\mathcal{O}}\left(\epsilon^{-2}\right), \gamma_t = \frac{\gamma}{\sqrt{t+1}}$ [14] & [Prop. 1] | $\Omega\left(\epsilon^{-2}\right)$ [11] | N/A due to lower bound | Nonconvergent $\forall$ bounded $\{\gamma_t\}$ [Thm. 3] |
| NSGD-M (Alg. 1) $\eta_t = \frac{\gamma}{(t+1)^\alpha\|g_t\|}$ | $\widetilde{\mathcal{O}}\left(\epsilon^{-2}\right), \alpha = 1/2$ [14] & [Prop. 1] | $\Omega\left(\epsilon^{-2}\right)$ [11] | $\widetilde{\mathcal{O}}\left(\epsilon^{-4}\right), \alpha = 3/4$ [14] & [Prop. 1] | $\Omega\left(\epsilon^{-4}\right)$ [3] |
| AMSGrad-norm (Alg. 2) $\eta_t = \frac{\gamma}{(t+1)^\alpha\sqrt{\hat{v}_{t+1}^2}}$ | $\widetilde{\mathcal{O}}\left(\epsilon^{-2/(1-\alpha)}\right), \alpha \in (0,1)$ [Thm. 5, 7] | $\Omega\left(\epsilon^{-2/(1-\alpha)}\right), \alpha \in (0,1)$ Nonconvergent, $\alpha = 0$ [Thm. 8] | N/A due to lower bound | $\Omega\left(\epsilon^{-2/(1-\zeta)}\right), \alpha = 1/2$ $\forall\zeta \in (0.5,1)$ [Thm. 4] |
| AdaGrad-norm (Alg. 5) $\eta_t = \frac{\eta}{\sqrt{v_0^2+\sum_{k=0}^t\|g(x_k;\xi_k)\|^2}}$ | $\widetilde{\mathcal{O}}\left(\epsilon^{-2}\right)$ [68] & [Prop. 3] | $\Omega\left(\epsilon^{-2}\right)$ [11] | $\widetilde{\mathcal{O}}\left(\epsilon^{-4}\right)$ [68] & [Prop. 3] | $\Omega\left(\epsilon^{-4}\right)$ [3] |

In this work, we study stochastic gradient methods for minimizing smooth functions in the parameter-agnostic regime. Firstly, we show SGD with polynomially decaying stepsize $1/\sqrt{t}$ is able to converge with the order-optimal rate, with and without bounded gradients (Proposition 4 and Theorem 1). Its limitation lies in an unavoidable exponential term in $\ell^2$ when we do not assume bounded gradients (Theorem 2). We demonstrate that several existing adaptive methods do not suffer from the exponential dependency, such as NGD, AdaGrad, AMSGrad-norm in the deterministic setting (Proposition 1 and Theorem 5), and NSGD-M, AdaGrad in the stochastic setting (Proposition 2 and Proposition 3). However, it does not mean adaptive methods are always better than SGD. We provide a non-convergence result for NSGD (Theorem 3) and a slow convergence result for AMSGrad-norm (Thoerem 4) in the stochastic case. We believe our results shed light on explaining commonly observed large gradients during training and provide a better theoretical understanding of the convergence behaviors of adaptive methods in the regime with unbounded stochastic gradients.

# B Proofs for SGD in Section 3

## B.1 Upper Bounds for SGD

We provide an extended theorem of Theorem 1 and include more general decaying stepsizes $\eta_t = \eta/(t+1)^\alpha$ with $0 < \alpha < 1$.

**Theorem 6.** *Under Assumptions 1 and 2, if we run SGD with stepsize $\eta_t = \eta/(t+1)^\alpha$ where $\eta > 0$ and $1/2 \le \alpha < 1$, then with $\eta \le 1/\ell$,*

$$\mathbb{E}\left[\frac{1}{T}\sum_{t=0}^{T-1}\|\nabla f(x_t)\|^2\right] \le \begin{cases} \frac{2}{\eta\sqrt{T}}\left(\Delta + \frac{\ell\sigma^2\eta^2}{2}(1+\log T)\right), & \text{when } \alpha = 1/2, \\ \frac{2}{\eta T^{1-\alpha}}\left(\Delta + \frac{\ell\sigma^2\eta^2}{2(1-2^{1-2\alpha})}\right), & \text{when } 1/2 < \alpha < 1, \\ \frac{2}{\eta T^\alpha}\left(\frac{\Delta}{T^{1-2\alpha}} + \frac{\ell\sigma^2\eta^2}{2(1-2\alpha)}\right), & \text{when } 0 < \alpha < 1/2; \end{cases}$$

*with $\eta > 1/\ell$,*

$$\mathbb{E}\left[\frac{1}{T}\sum_{t=0}^{T-1}\|\nabla f(x_t)\|^2\right] \leq \begin{cases} \frac{\sqrt{2}\,(4e)^\tau}{\eta\sqrt{\pi\tau T}}\left[1 + \ell\eta\left(1 + 2\sqrt{\tau}\right)\right]\left(\Delta + \frac{\ell\sigma^2\eta^2}{2}(1 + \log T)\right), \\ \qquad \textit{when } \alpha = 1/2, \\ \frac{2\,(4e^{2\alpha})^\tau}{\eta(2\pi\tau)^\alpha\,T^{1-\alpha}}\left[1 + \ell\eta\left(1 + \frac{\tau^{1-\alpha}}{1-\alpha}\right)\right]\left(\Delta + \frac{\ell\sigma^2\eta^2}{2\left(1 - 2^{1-2\alpha}\right)}\right), \\ \qquad \textit{when } 1/2 < \alpha < 1, \\ \frac{2\,(4e^{2\alpha})^\tau}{\eta(2\pi\tau)^\alpha\,T^\alpha}\left[1 + \ell\eta\left(1 + \frac{\tau^{1-\alpha}}{1-\alpha}\right)\right]\left(\frac{\Delta}{T^{1-2\alpha}} + \frac{\ell\sigma^2\eta^2}{2\left(1 - 2\alpha\right)}\right), \\ \qquad \textit{when } 0 < \alpha < 1/2, \end{cases}$$

*where $\tau = \lceil (\eta\ell)^{1/\alpha} - 1 \rceil$.*

**Proof.** By $\ell$-smoothness of $f(\cdot)$,

$$f(x_{t+1}) \leq f(x_t) + \langle \nabla f(x_t), x_{t+1} - x_t \rangle + \frac{\ell}{2}\|x_{t+1} - x_t\|^2$$

$$= f(x_t) - \eta_t\langle \nabla f(x_t), g(x_t; \xi_t)\rangle + \frac{\ell\eta_t^2}{2}\|g(x_t; \xi_t)\|^2$$

Taking expectation,

$$\mathbb{E}f(x_{t+1}) \leq \mathbb{E}f(x_t) - \eta_t\mathbb{E}\|\nabla f(x_t)\|^2 + \frac{\ell\eta_t^2}{2}\mathbb{E}\|\nabla f(x_t)\|^2 + \frac{\ell\eta_t^2}{2}\sigma^2$$

$$\leq \mathbb{E}f(x_t) - \left(\eta_t - \frac{\ell\eta_t^2}{2}\right)\mathbb{E}\|\nabla f(x_t)\|^2 + \frac{\ell\eta_t^2}{2}\sigma^2 \qquad (2)$$

We note that $\eta_t - \frac{\ell\eta_t^2}{2} \geq \frac{\eta_t}{2}$ when $\eta_t \leq \frac{1}{\ell}$, i.e., $t \geq (\eta\ell)^{1/\alpha} - 1$. Define $\tau = \lceil (\eta\ell)^{1/\alpha} - 1 \rceil$. Therefore, for all $t < \tau$,

$$\mathbb{E}f(x_{t+1}) \leq \mathbb{E}f(x_t) + \frac{\ell\eta_t^2}{2}\mathbb{E}\|\nabla f(x_t)\|^2 + \frac{\ell\eta_t^2}{2}\sigma^2. \qquad (3)$$

For all $t \geq \tau$, we have

$$\mathbb{E}f(x_{t+1}) \leq \mathbb{E}f(x_t) - \frac{\eta_t}{2}\mathbb{E}\|\nabla f(x_t)\|^2 + \frac{\ell\eta_t^2}{2}\sigma^2. \qquad (4)$$

Summing from $t = \tau$ to $T - 1$, we have

$$\sum_{t=\tau}^{T-1}\frac{\eta_t}{2}\mathbb{E}\|\nabla f(x_t)\|^2 \leq \mathbb{E}f(x_\tau) - \mathbb{E}f(x_T) + \sum_{t=\tau}^{T-1}\frac{\ell\eta_t^2}{2}\sigma^2 \qquad (5)$$

Now we want to bound $\mathbb{E}f(x_\tau) - f(x_T) \leq \mathbb{E}f(x_\tau) - f^*$, where $f^* \triangleq \min_{x\in\mathbb{R}^d} f(x)$. From (3),

$$\mathbb{E}f(x_{t+1}) - f^* \leq \mathbb{E}f(x_t) - f^* + \frac{\ell\eta_t^2}{2}\mathbb{E}\|\nabla f(x_t)\|^2 + \frac{\ell\eta_t^2}{2}\sigma^2$$

$$\leq (1 + \ell^2\eta_t^2)[\mathbb{E}f(x_t) - f^*] + \frac{\ell\eta_t^2}{2}\sigma^2,$$

where in the second inequality we use $\|\nabla f(x)\|^2 \leq 2\ell[f(x) - f^*]$. When $\tau = 0$, $f(x_\tau) - f(x_T) \leq \Delta$; when $\tau \geq 1$, recursing the inequality above, for $j \leq \tau$,

$$\mathbb{E}f(x_j) - f^* \leq \Delta\left(\prod_{t=0}^{j-1} 1 + \ell^2\eta_t^2\right) + \sum_{k=0}^{j-2}\left(\prod_{t=k+1}^{j-1} 1 + \ell^2\eta_t^2\right)\frac{\ell\eta_k^2}{2}\sigma^2 + \frac{\ell\eta_{j-1}^2}{2}\sigma^2$$

$$\leq \left(\prod_{t=0}^{j-1} 1 + \ell^2 \eta_t^2\right)\left(\Delta + \sum_{t=0}^{j-1} \frac{\ell \eta_t^2}{2}\sigma^2\right)$$

$$\leq \left(\prod_{t=0}^{\tau-1} 1 + \ell^2 \eta_t^2\right)\left(\Delta + \sum_{t=0}^{\tau-1} \frac{\ell \eta_t^2}{2}\sigma^2\right). \tag{6}$$

Also, with $\|\nabla f(x)\|^2 \leq 2\ell[f(x) - f^*]$, if $\tau \geq 1$,

$$\sum_{t=0}^{\tau-1} \frac{\eta_t}{2}\mathbb{E}\|\nabla f(x_t)\|^2 \leq \sum_{t=0}^{\tau-1} \eta_t \ell \mathbb{E}\left(f(x_t) - f^*\right)$$

$$\leq \ell\left(\sum_{t=0}^{\tau-1} \eta_t\right)\left(\prod_{t=0}^{\tau-1} 1 + \ell^2 \eta_t^2\right)\left(\Delta + \sum_{t=0}^{\tau-1} \frac{\ell \eta_t^2}{2}\sigma^2\right),$$

where in the second inequality we use (6) . Combining with (5) and (6), if $\tau \geq 1$

$$\sum_{t=0}^{T-1} \frac{\eta_t}{2}\mathbb{E}\|\nabla f(x_t)\|^2 \leq \left(\prod_{t=0}^{\tau-1} 1 + \ell^2 \eta_t^2\right)\left(\Delta + \sum_{t=0}^{\tau-1} \frac{\ell \eta_t^2}{2}\sigma^2\right) + \sum_{t=0}^{T-1} \frac{\ell \eta_t^2}{2}\sigma^2$$

$$+ \ell\left(\sum_{t=0}^{\tau-1} \eta_t\right)\left(\prod_{t=0}^{\tau-1} 1 + \ell^2 \eta_t^2\right)\left(\Delta + \sum_{t=0}^{\tau-1} \frac{\ell \eta_t^2}{2}\sigma^2\right).$$

We note that

$$\prod_{t=0}^{\tau-1}\left(1 + \ell^2 \eta_t^2\right) = \prod_{t=0}^{\tau-1}\left(1 + \frac{\ell^2 \eta^2}{(t+1)^{2\alpha}}\right) = \frac{\prod_{t=0}^{\tau-1}\left(\ell^2\eta^2 + (t+1)^{2\alpha}\right)}{(\tau!)^{2\alpha}} \leq \frac{\left(\ell^2\eta^2 + \tau^{2\alpha}\right)^\tau}{(\tau!)^{2\alpha}}$$

$$\leq \frac{\left(2\ell^2\eta^2\right)^\tau}{(\tau!)^{2\alpha}} \leq \frac{(2\ell^2\eta^2)^\tau}{\left[\sqrt{2\pi\tau}\left(\frac{\tau}{e}\right)^\tau \exp\left(\frac{1}{12\tau+1}\right)\right]^{2\alpha}}$$

$$\leq \frac{1}{(2\pi\tau)^\alpha}\left(\frac{2\ell^2\eta^2 e^{2\alpha}}{\tau^{2\alpha}}\right)^\tau \leq \frac{1}{(2\pi\tau)^\alpha}\left(4e^{2\alpha}\right)^\tau,$$

where in the third inequality we use Stirling's approximation. Therefore,

$$\sum_{t=0}^{T-1} \frac{\eta_t}{2}\mathbb{E}\|\nabla f(x_t)\|^2 \leq \frac{1}{(2\pi\tau)^\alpha}\left(4e^{2\alpha}\right)^\tau\left[1 + \ell\left(\sum_{t=0}^{\tau-1} \eta_t\right)\right]\left(\Delta + \sum_{t=0}^{T-1} \frac{\ell \eta_t^2}{2}\sigma^2\right).$$

Plugging in $\eta_t = \eta/(t+1)^\alpha$, when $\alpha = 1/2$,

$$\sum_{t=0}^{T-1} \mathbb{E}\|\nabla f(x_t)\|^2 \leq \frac{\sqrt{2T}}{\eta\sqrt{\pi\tau}}(4e)^\tau\left[1 + \ell\eta\left(1 + 2\sqrt{\tau}\right)\right]\left(\Delta + \frac{\ell\sigma^2\eta^2}{2}(1 + \log T)\right);$$

when $1/2 < \alpha < 1$,

$$\sum_{t=0}^{T-1} \mathbb{E}\|\nabla f(x_t)\|^2 \leq \frac{2\,T^\alpha}{\eta(2\pi\tau)^\alpha}(4e^{2\alpha})^\tau\left[1 + \ell\eta\left(1 + \frac{\tau^{1-\alpha}}{1-\alpha}\right)\right]\left(\Delta + \frac{\ell\sigma^2\eta^2}{2\left(1 - 2^{1-2\alpha}\right)}\right).$$

when $0 < \alpha < 1/2$,

$$\sum_{t=0}^{T-1} \mathbb{E}\|\nabla f(x_t)\|^2 \leq \frac{2\,T^\alpha}{\eta(2\pi\tau)^\alpha}(4e^{2\alpha})^\tau\left[1 + \ell\eta\left(1 + \frac{\tau^{1-\alpha}}{1-\alpha}\right)\right]\left(\Delta + \frac{\ell\sigma^2\eta^2 T^{1-2\alpha}}{2\left(1 - 2\alpha\right)}\right);$$

If $\tau = 0$, from (5),

$$\sum_{t=0}^{T-1} \frac{\eta_t}{2} \mathbb{E}\|\nabla f(x_t)\|^2 \leq \Delta + \sum_{t=0}^{T-1} \frac{\ell \eta_t^2}{2} \sigma^2,$$

Plugging in $\eta_t$, when $\alpha = 1/2$,

$$\sum_{t=0}^{T-1} \mathbb{E}\|\nabla f(x_t)\|^2 \leq \frac{2\sqrt{T}}{\eta} \left( \Delta + \frac{\ell\sigma^2\eta^2}{2}(1 + \log T) \right);$$

when $1/2 < \alpha < 1$,

$$\sum_{t=0}^{T-1} \mathbb{E}\|\nabla f(x_t)\|^2 \leq \frac{2T^\alpha}{\eta} \left( \Delta + \frac{\ell\sigma^2\eta^2}{2\left(1 - 2^{1-2\alpha}\right)} \right).$$

when $0 < \alpha < 1/2$,

$$\sum_{t=0}^{T-1} \mathbb{E}\|\nabla f(x_t)\|^2 \leq \frac{2T^\alpha}{\eta} \left( \Delta + \frac{\ell\sigma^2\eta^2 T^{1-2\alpha}}{2\left(1 - 2\alpha\right)} \right).$$

$\square$

**Remark 2.** *When we run SGD with stepsize $\eta_t = \eta/(t+1)^\alpha$, where $1/2 < \alpha < 1$, Theorem 6 implies a complexity of $\mathcal{O}\left( (4e^{2\alpha})^{\frac{(\eta\ell)^{1/\alpha}}{1-\alpha}} (\eta\ell)^{\frac{1}{\alpha(1-\alpha)}} \cdot \epsilon^{\frac{-2}{1-\alpha}} \right)$ in the large initial stepsize regime $\eta > 1/\ell$. Compared with the case $\alpha = 1/2$, when $\alpha$ is larger, the convergence rate in $T$ is slower, but it also comes with a smaller exponent, i.e., $(\eta\ell)^{1/\alpha}$. This is because $\alpha = 1/2$ leads to the best convergence rate in $T$ [17], while the faster decaying stepsize $\alpha > 1/2$ will reach the desirable stepsize $1/\ell$ earlier so that it accumulates less gradient norms before $\tau$. For $0 < \alpha < 1/2$, however, it comes with both a larger exponent and a slower convergence rate.*

**Proposition 4** (with bounded gradient)**.** *Under Assumption 1, 2 and additionally assuming that the gradient norm is upper bounded by $G$, i.e., $\|\nabla f(x)\| \leq G$ for all $x \in \mathbb{R}^d$, if we run SGD with stepsize $\eta_t = \eta/\sqrt{t+1}$ with $\eta > 0$, then*

$$\mathbb{E}\left[ \frac{1}{T} \sum_{t=0}^{T-1} \|\nabla f(x_t)\|^2 \right] \leq \frac{1}{\sqrt{T}} \left( \frac{\Delta}{\eta} + \frac{\ell\eta\left(G^2 + \sigma^2\right)}{2} \log T \right).$$

**Proof.** By the smoothness of $f(\cdot)$, we have

$$f(x_{t+1}) \leq f(x_t) - \eta_t \langle \nabla F(x_t; \xi_t), \nabla f(x_t) \rangle + \frac{\ell \eta_t^2}{2} \|\nabla F(x_t; \xi_t)\|^2.$$

Taking expectation and summing from $t = 0$ to $T - 1$,

$$\mathbb{E}\left[ \sum_{t=0}^{T-1} \eta_t \|\nabla f(x_t)\|^2 \right] \leq f(x_0) - f(x_T) + \frac{\ell}{2} \sum_{t=0}^{T-1} \eta_t^2 \mathbb{E}\left[ \|\nabla F(x_t; \xi_t)\|^2 \right]$$

$$\leq \Delta + \frac{\ell}{2} \sum_{t=0}^{T-1} \eta_t^2 \mathbb{E}\left[ \|\nabla F(x_t; \xi_t)\|^2 \right]$$

$$\leq \Delta + \frac{\ell}{2} \sum_{t=0}^{T-1} \eta_t^2 \left( \|\nabla f(x_t)\|^2 + \mathbb{E}\left[ \|\nabla F(x_t; \xi_t) - f(x_t)\|^2 \right] \right)$$

$$\leq \Delta + \frac{\ell}{2} \sum_{t=0}^{T-1} \eta_t^2 \left( G^2 + \sigma^2 \right).$$

Let $\eta_t = \eta/\sqrt{t+1}$,

$$\frac{\eta}{\sqrt{T}}\mathbb{E}\left[\sum_{t=0}^{T-1}\|\nabla f(x_t)\|^2\right] \le \mathbb{E}\left[\sum_{t=0}^{T-1}\frac{\eta}{\sqrt{t+1}}\|\nabla f(x_t)\|^2\right] \le \Delta + \frac{\ell\left(G^2+\sigma^2\right)}{2}\sum_{t=0}^{T-1}\frac{\eta^2}{t+1},$$

$$\le \Delta + \frac{\ell\eta^2\left(G^2+\sigma^2\right)}{2}\log T.$$

$\square$

## B.2 Lower Bound for SGD

**Proof for Theorem 2**

**Proof.** We construct the hard instance with 4 segments of quadratic functions. The function is symmetric about $x = 0$, and we will define it on $x \le 0$ as below. We illustrate it in Figure 2.

*Segment 1.* We define $f(x) = \frac{\ell}{2}x^2$. We pick $x_0$ such that $f(x_0) - f^* = \Delta$, i.e., $x_0 = \sqrt{\frac{2\Delta}{\ell}}$. We define $t_0$ to be the first iteration that $\eta_{t_0} = \frac{\eta}{\sqrt{t_0+1}} \ge \frac{4}{\ell}$, i.e., $t_0 = \left\lfloor \frac{\eta^2\ell^2}{16} - 1 \right\rfloor$. With the update rule $x_{t+1} = x_t - \frac{\eta\ell}{\sqrt{t+1}}x_t = \left(1 - \frac{\eta\ell}{\sqrt{t+1}}\right)x_t$, we have for $t \le t_0$

$$|x_t|^2 = \left[\prod_{k=1}^{t}\left(\frac{\eta\ell}{\sqrt{k}}-1\right)\right]^2 |x_0|^2 \ge \prod_{k=1}^{t}\frac{\eta^2\ell^2}{2k}|x_0|^2$$

$$= \frac{(\eta^2\ell^2/2)^t}{(t)!}|x_0|^2 > \frac{(\eta^2\ell^2/2)^t}{\sqrt{2\pi t}(t/e)^t e^{1/12t}}|x_0|^2 \ge \frac{1}{3\sqrt{t}}(8e)^t|x_0|^2,$$

where in the inequality we use $\left(\frac{\eta\ell}{\sqrt{k}}-1\right)^2 \ge \frac{\eta^2\ell^2}{2k}$ with $k \le t_0$, in the second inequality we use Stirling's approximation, and in the last inequality we use $t \le \eta^2\ell^2/16$. We note that

$$|x_{t_0}|^2 \ge \frac{1}{3\sqrt{t_0}}(8e)^{t_0}|x_0|^2 \ge \frac{4}{3\eta\ell}(8e)^{\eta^2\ell^2/16-2}|x_0|^2.$$

Without loss of generality, we assume $x_{t_0} > 0$. Segment 1 is define on the domain $\{x : |x| \le x_{t_0}\}$.

*Segment 2.* This segment is the mirror of Segment 1. On domain $\{x : x_{t_0} \le x \le 2x_{t_0}\}$, we define $f(x) = -\frac{\ell}{2}(x - 2x_{t_0})^2 + \ell x_{t_0}^2$.

*Segment 3.* We note that

$$x_{t_0+1} = x_{t_0} - \eta_{t_0}\ell x_{t_0} = \left(1 - \frac{\eta\ell}{\sqrt{t_0+1}}\right)x_{t_0} \le -3x_{t_0},$$

where the inequality is from the definition of $t_0$, and

$$\tilde{\Delta} \triangleq \frac{\ell x_{t_0}^2}{2} \ge \frac{2}{3\eta}(8e)^{\eta^2\ell^2/16-2}|x_0|^2 = \frac{4}{3\eta\ell}(8e)^{\eta^2\ell^2/16-2}\Delta.$$

We construct a quadratic function such that: it passes $(-2x_{t_0}, \ell x_{t_0}^2)$ with gradient 0; the gradient at $x = x_{t_0+1}$ is $\frac{\sqrt{\tilde{\Delta}}}{2\sqrt{\max\{1/\ell,\sum_{t=t_0+1}^{T-1}\eta_t\}}}$. This quadratic function is uniquely defined to be

$$f(x) = -\frac{\sqrt{\tilde{\Delta}}\,(x+2x_{t_0})^2}{4(-2x_{t_0}-x_{t_0+1})\sqrt{\max\{1/\ell,\sum_{t=t_0+1}^{T-1}\eta_t\}}} + \ell x_{t_0}^2.$$

It can be verified that this function is $\ell$-smooth: as $x_{t_0+1} \leq -3x_{t_0}$,

$$\frac{\sqrt{\frac{1}{2}\ell x_{t_0}^2}}{2(-2x_{t_0} - x_{t_0+1})} \leq \sqrt{\ell} \iff \frac{\sqrt{\tilde{\Delta}}}{2(-2x_{t_0} - x_{t_0+1})\sqrt{\ell}} \leq \ell$$

$$\implies \frac{\sqrt{\tilde{\Delta}}}{2(-2x_{t_0} - x_{t_0+1})\sqrt{\max\{1/\ell, \sum_{t=t_0+1}^{T-1} \eta_t\}}} \leq \ell.$$

The function is defined on the domain $\{x : x_{t_0+1} \leq x \leq -2x_{t_0}\}$.

*Segment 4.* For convenience, we define $w = f(x_{t_0+1})$. We can verify that $w \geq \tilde{\Delta}$: as $\frac{1}{\sqrt{t_0}} < \frac{4}{\eta\ell}$,

$$-2x_{t_0} - \left(1 - \frac{\eta\ell}{\sqrt{t_0+1}}\right) x_{t_0} \leq 4\sqrt{\frac{x_{t_0}}{2}} \iff \frac{-2x_{t_0} - x_{t_0+1}}{4} \leq \sqrt{\frac{x_{t_0}}{2}}$$

$$\implies \frac{-2x_{t_0} - x_{t_0+1}}{4\sqrt{\max\{1/\ell, \sum_{t=t_0+1}^{T-1} \eta_t\}}} \leq \sqrt{\frac{1}{2}\ell x_{t_0}^2} \iff \frac{\sqrt{\tilde{\Delta}}\,(x_{t_0+1} + 2x_{t_0})^2}{4(-2x_{t_0} - x_{t_0+1})\sqrt{\max\{1/\ell, \sum_{t=t_0+1}^{T-1} \eta_t\}}} \leq \tilde{\Delta}.$$

So we conclude $w \geq \tilde{\Delta}$. Now we construct a quadratic function similar to that in Proposition 1 of [17]: it passes $(x_{t_0+1}, w)$ with gradient $\dfrac{\sqrt{\tilde{\Delta}}}{2\sqrt{\max\{1/\ell, \sum_{t=t_0+1}^{T-1} \eta_t\}}}$; the minimum is at $x = x_{t_0+1} - \sqrt{\tilde{\Delta}\max\{1/\ell, \sum_{t=t_0+1}^{T-1} \eta_t\}}$. This quadratic function is defined to be

$$f(x) = \frac{\left(x - x_{t_0+1} + \sqrt{\tilde{\Delta}\max\{1/\ell, \sum_{t=t_0+1}^{T-1} \eta_t\}}\right)^2}{4\max\{1/\ell, \sum_{t=t_0+1}^{T-1} \eta_t\}} + w - \frac{\tilde{\Delta}}{4}$$

on the domain $\{x : x \leq x_{t_0+1}\}$. It is obvious that $f(x) \geq 0$ and is $\ell$-smooth. Following the same reasoning of Proposition 1 in [17], also presented as Lemma 1 in the appendix for completeness, we can conclude for all $t : t_0 + 1 \leq t \leq T$,

$$|\nabla f(x_t)| \geq \frac{\sqrt{\tilde{\Delta}}}{4\sqrt{\max\{1/\ell, \sum_{t=t_0+1}^{T-1} \eta_t\}}} \geq \frac{1}{4}\sqrt{\tilde{\Delta}} \min\left\{\sqrt{\ell}, (2\eta)^{-1/2}T^{-1/4}\right\},$$

where in the second inequality we use $\sum_{t=t_0+1}^{T-1} \eta_t = \sum_{t=t_0+1}^{T-1} \frac{\eta}{\sqrt{t+1}} \leq 2\eta T^{1/2}$.

$\square$

The following lemma is used in the proof of Theorem 2. It is a straightforward modification of Proposition 1 in [17]. We present it here for completeness.

**Lemma 1.** *Under the same setting and notations as the proof of Theorem 2, if we run gradient descent with stepsize $\{\eta_t\}_{t=t_0+1}^{T-1}$ starting from point $x_{t_0+1}$ on function*

$$f(x) = \frac{\left(x - x_{t_0+1} + \sqrt{\tilde{\Delta}\max\{1/\ell, \sum_{t=t_0+1}^{T-1} \eta_t\}}\right)^2}{4\max\{1/\ell, \sum_{t=t_0+1}^{T-1} \eta_t\}} + w - \frac{\tilde{\Delta}}{4},$$

*then for all $t : t_0 + 1 \leq t \leq T$,*

$$|\nabla f(x_t)| \geq \frac{\sqrt{\tilde{\Delta}}}{4\sqrt{\max\{1/\ell, \sum_{t=t_0+1}^{T-1} \eta_t\}}}.$$

**Proof.** From the update of gradient descent, we have

$$x_{t+1} = x_t - \eta_t \cdot \frac{x_t - x_{t_0+1} + \sqrt{\tilde{\Delta} \max\{1/\ell, \sum_{t=t_0+1}^{T-1} \eta_t\}}}{2 \max\{1/\ell, \sum_{t=t_0+1}^{T-1} \eta_t\}},$$

which leads to

$$x_{t+1} - x_{t_0+1} + \sqrt{\tilde{\Delta} \max\{1/\ell, \sum_{t=t_0+1}^{T-1} \eta_t\}}$$

$$= \left(1 - \frac{\eta_t}{2 \max\{1/\ell, \sum_{t=t_0+1}^{T-1} \eta_t\}}\right) \left(x_t - x_{t_0+1} + \sqrt{\tilde{\Delta} \max\{1/\ell, \sum_{t=t_0+1}^{T-1} \eta_t\}}\right).$$

Recursing this, for $j \leq T$

$$x_j - x_{t_0+1} + \sqrt{\tilde{\Delta} \max\{1/\ell, \sum_{t=t_0+1}^{T-1} \eta_t\}}$$

$$= \prod_{k=t_0+1}^{j-1} \left(1 - \frac{\eta_k}{2 \max\{1/\ell, \sum_{t=t_0+1}^{T-1} \eta_t\}}\right) \left(x_{t_0+1} - x_{t_0+1} + \sqrt{\tilde{\Delta} \max\{1/\ell, \sum_{t=t_0+1}^{T-1} \eta_t\}}\right)$$

$$\geq \exp\left(\log \frac{1}{2} \cdot \sum_{k=t_0+1}^{j-1} \frac{\eta_k}{2 \max\{1/\ell, \sum_{t=t_0+1}^{T-1} \eta_t\}}\right) \sqrt{\tilde{\Delta} \max\{1/\ell, \sum_{t=t_0+1}^{T-1} \eta_t\}}$$

$$\geq \frac{1}{2} \sqrt{\tilde{\Delta} \max\{1/\ell, \sum_{t=t_0+1}^{T-1} \eta_t\}},$$

where in the second inequality, we use $1 - z/2 \geq \exp(\log \frac{1}{2} \cdot z)$ for $0 \leq z \leq 1$. This directly implies what we want to prove by computing $\nabla f(x_j)$.

$\square$

## C  Proofs for NSGD Family in Section 4

---
**Algorithm 3** Normalized Stochastic Gradient Descent (NSGD)
---
1: **Input:** initial point $x_0$
2: **for** $t = 0, 1, 2, \dots$ **do**
3:    sample $\xi_t$ and set learning rate $\gamma_t$
4:    $x_{t+1} = x_t - \frac{\gamma_t}{\|g(x_t;\xi_t)\|} g(x_t;\xi_t)$
5: **end for**
---

**Proof for Theorem 3**

**Proof.** Let us pick $f(x) = \frac{L}{2}x^2$ with $\frac{\epsilon^2}{2\Delta} < L \leq \ell$ and $L < \frac{\sigma - \epsilon}{\gamma_{\max}}$. Then we pick $x_0$ such that $\frac{\epsilon}{L} < x_0 < \sqrt{\frac{2\Delta}{L}}$, which implies that $\|\nabla f(x_0)\| > \epsilon$ and $f(x_0) - \min_x f \leq \Delta$. Now we define $D = \{x : -w \leq x \leq w\}$ with $\frac{\epsilon}{L} + \gamma_{\max} < w < \frac{\sigma}{L}$. For $x \in D$, we have $\|\nabla f(x)\| \leq \sigma$ and we

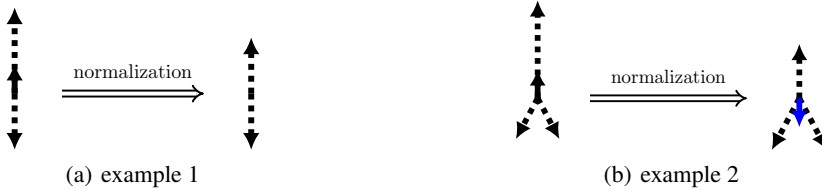

|  (a) example 1 |  (b) example 2 |

Figure 3: The expected update of NSGD can vanish (example 1) or be in the opposite direction (example 2) of the true gradient. The solid black arrow represents the true gradient and the dashed arrows are the possible stochastic gradients (with equal possibilities). The solid blue arrow is the expected direction of NSGD update.

construct the noisy gradients: with $\delta > 1$

$$g(x;\xi) = (1+\delta)\nabla f(x) \text{ w.p. } \frac{1}{2}, \text{ and } g(x;\xi) = (1-\delta)\nabla f(x) \text{ w.p. } \frac{1}{2}.$$

It is obvious that $\nabla f(x) = \mathbb{E}[g(x;\xi)]$ and the variance at this point $\mathbb{E}\|\nabla f(x) - g(x;\xi)\|^2 = \delta^2\|\nabla f(x)\|^2 \leq \sigma^2$ with $\delta$ sufficiently close to 1. With the update rule, we note that $x_{t+1} = x_t - \gamma_t$ w.p. 1/2 and $x_{t+1} = x_t + \gamma_t$ w.p. 1/2, and therefore

$$\mathbb{E}_{\xi_t}\left[\|\nabla f(x_{t+1})|x_t \in D\|\right] = \frac{1}{2}[L\|x_t - \gamma_t\| + L\|x_t + \gamma_t\|] \geq L\|x_t\| = \|\nabla f(x_t)\|.$$

For $x \notin D$, we have $\|x\| > \epsilon/L + \gamma_{\max}$, and we assume there is no noise in the gradients. Therefore, if $x_t \notin D$, we know that after one step of update $\|x_{t+1}\| > \epsilon/L$, which implies $\|\nabla f(x_{t+1})\| > \epsilon$. Combining two cases that $x_t \in D$ and $x_t \notin D$, we know that $\mathbb{E}\|\nabla f(x_t)\| > \epsilon$ for all $t$.

$\square$

**Proof of Proposition 1**

**Proof.** Denote $e_t = g(x_t;\xi_t) - \nabla f(x_t)$. By Lemma 2 in [14],

$$f(x_{t+1}) - f(x_t) \leq -\frac{\gamma_t}{3}\|\nabla f(x_t)\| + \frac{8\gamma_t}{3}\|e_t\| + \frac{\ell\gamma_t^2}{2}.$$

Telescoping from $t = 0$ to $T - 1$,

$$\frac{\gamma}{3T^{1/2}}\sum_{t=0}^{T-1}\|\nabla f(x_t)\| \leq \frac{1}{3}\sum_{t=0}^{T-1}\gamma_t\|\nabla f(x_t)\| \leq \Delta + \frac{8}{3}\sum_{t=0}^{T-1}\gamma_t\|e_t\| + \sum_{t=0}^{T-1}\frac{\ell\gamma_t^2}{2},$$

Taking expectation, rearranging and using $\mathbb{E}[\|e_t\|] \leq \left(\mathbb{E}[\|e_t\|^2]\right)^{1/2} \leq \sigma$, we derive

$$\sum_{t=0}^{T-1}\mathbb{E}[\|\nabla f(x_t)\|] \leq 3T^{1/2}\left[\frac{\Delta}{\gamma} + \frac{8\sigma}{3}\sum_{t=0}^{T-1}\frac{1}{(t+1)^{1/2}} + \frac{\ell\gamma}{2}\sum_{t=0}^{T-1}\frac{1}{t+1}\right]$$

$$\leq 3T^{1/2}\left[\frac{\Delta}{\gamma} + 8\sigma T^{1/2} + \ell\gamma\log(T)\right].$$

$\square$

**Proof for Proposition 2**

**Proof.** We define $\hat{e}_t = g_t - \nabla f(x_t)$. By Lemma 2 in [14], for any $\gamma_t > 0$

$$f(x_{t+1}) - f(x_t) \leq -\frac{\gamma_t}{3}\|\nabla f(x_t)\| + \frac{8\gamma_t}{3}\|\hat{e}_t\| + \frac{\ell\gamma_t^2}{2}. \tag{7}$$

Telescoping from $t = 0$ to $T - 1$,

$$\frac{\gamma}{3T^{3/4}}\sum_{t=0}^{T-1}\|\nabla f(x_t)\| \leq \frac{1}{3}\sum_{t=0}^{T-1}\gamma_t\|\nabla f(x_t)\| \leq \Delta + \frac{8}{3}\sum_{t=0}^{T-1}\gamma_t\|\hat{e}_t\| + \sum_{t=0}^{T-1}\frac{\ell\gamma_t^2}{2},$$

By taking expectation on both sides, rearranging and controlling the variance term using Lemma 2, we derive

$$\begin{aligned}
\sum_{t=0}^{T-1}\mathbb{E}\left[\|\nabla f(x_t)\|\right] &\leq 3T^{3/4}\left[\frac{\Delta}{\gamma} + \frac{8}{3\gamma}\sum_{t=0}^{T-1}\gamma_t\mathbb{E}\left[\|\hat{e}_t\|\right] + \frac{\ell\gamma}{2}\sum_{t=0}^{T-1}(t+1)^{-3/2}\right] \\
&\leq 3T^{3/4}\left[\frac{\Delta}{\gamma} + \frac{8}{3}\left(C_1\sigma + C_2\ell\gamma\right)\log(T) + \frac{2\ell\gamma}{T^{1/2}}\right] \\
&\leq CT^{3/4}\left[\frac{\Delta}{\gamma} + (\sigma + \ell\gamma)\log(T)\right].
\end{aligned}$$

$\square$

**Lemma 2.** *Under the setting of Theorem 2, there exist numerical constants $C_1, C_2 > 0$ such that for all $t \geq 1$,*

$$\mathbb{E}\left[\|\hat{e}_t\|\right] \leq C_1\sigma\alpha_t^{1/2} + C_2\ell\gamma_t\alpha_t^{-1},$$

$$\sum_{t=0}^{T-1}\gamma_t\mathbb{E}\left[\|\hat{e}_t\|\right] \leq \left(C_1\sigma\gamma + C_2\ell\gamma^2\right)\log(T),$$

*where $\hat{e}_t = g_t - \nabla f(x_t)$.*

**Proof.** Define $e_t = g(x_t; \xi_t) - \nabla f(x_t)$, $S_t = \nabla f(x_t) - \nabla f(x_{t+1})$. Then

$$\begin{aligned}
\hat{e}_{t+1} &= g_{t+1} - \nabla f(x_{t+1}) \\
&= (1 - \alpha_t)g_t + \alpha_t g(x_{t+1}; \xi_{t+1}) - \nabla f(x_{t+1}) \\
&= (1 - \alpha_t)\hat{e}_t + \alpha_t\epsilon_{t+1} + (1 - \alpha_t)S_t.
\end{aligned}$$

Unrolling the recursion from $t = T - 1$ to $t = 0$, we have

$$\hat{e}_T = \left(\prod_{t=0}^{T-1}(1 - \alpha_t)\right)\hat{e}_0 + \sum_{t=0}^{T-1}\alpha_t e_{t+1}\prod_{\tau=t+1}^{T-1}(1 - \alpha_\tau) + \sum_{t=0}^{T-1}(1 - \alpha_t)S_t\prod_{\tau=t+1}^{T-1}(1 - \alpha_\tau). \tag{8}$$

Define the $\sigma$-field $\mathcal{F}_t := \sigma(\{x_0, \xi_0, \ldots, \xi_{t-1}\})$. Notice that for any $t_2 > t_1 \geq 0$ we have

$$\mathbb{E}\left[\langle e_{t_1}, e_{t_2}\rangle\right] = \mathbb{E}\left[\mathbb{E}\left[\langle e_{t_1}, e_{t_2}\rangle|\mathcal{F}_{t_2}\right]\right] = \mathbb{E}\left[\langle e_{t_1}, \mathbb{E}\left[e_{t_2}|\mathcal{F}_{t_2}\right]\rangle\right] = 0. \tag{9}$$

Then taking norm, applying expectation on both sides of (8) and using $\mathbb{E}\left[\|\hat{e}_0\|\right] \leq \sigma$, we have

$$\begin{aligned}
\mathbb{E}\left[\|\hat{e}_T\|\right] &\leq \left(\prod_{t=0}^{T-1}(1 - \alpha_t)\right)\sigma + \mathbb{E}\left[\left\|\sum_{t=0}^{T-1}\alpha_t e_{t+1}\prod_{\tau=t+1}^{T-1}(1 - \alpha_\tau)\right\|\right] \\
&\quad + \mathbb{E}\left[\left\|\sum_{t=0}^{T-1}(1 - \alpha_t)S_t\prod_{\tau=t+1}^{T-1}(1 - \alpha_\tau)\right\|\right]
\end{aligned}$$

$$\leq \left(\prod_{t=0}^{T-1}(1-\alpha_t)\right)\sigma + \left(\mathbb{E}\left[\left\|\sum_{t=0}^{T-1}\alpha_t e_{t+1}\prod_{\tau=t+1}^{T-1}(1-\alpha_\tau)\right\|^2\right]\right)^{1/2}$$

$$+ \sum_{t=0}^{T-1}(1-\alpha_t)\mathbb{E}\left[\|S_t\|\right]\prod_{\tau=t+1}^{T-1}(1-\alpha_\tau)$$

$$\leq \left(\prod_{t=0}^{T-1}(1-\alpha_t)\right)\sigma + \left(\sum_{t=0}^{T-1}\alpha_t^2\mathbb{E}\left[\|e_{t+1}\|^2\right]\prod_{\tau=t+1}^{T-1}(1-\alpha_\tau)^2\right)^{1/2}$$

$$+\ell\sum_{t=0}^{T-1}(1-\alpha_t)\gamma_t\prod_{\tau=t+1}^{T-1}(1-\alpha_\tau)$$

$$\leq \left(\prod_{t=0}^{T-1}(1-\alpha_t)\right)\sigma + \left(\sum_{t=0}^{T-1}\alpha_t^2\prod_{\tau=t+1}^{T-1}(1-\alpha_\tau)\right)^{1/2}\sigma + \left(\sum_{t=0}^{T-1}\gamma_t\prod_{\tau=t+1}^{T-1}(1-\alpha_\tau)\right)\ell,$$

where the first inequality holds by Jensen's inequality applied to $x \mapsto x^2$, the second inequality follows by (9) and the bound $\|S_t\| \leq \ell\|x_{t+1} - x_t\| = \ell\gamma_t$. The last step is due to bounded variance $\mathbb{E}\left[\|\hat\epsilon_0\|\right] \leq \sigma$ and $\alpha_t \leq 1$.

By the choice of momentum sequence, we have $\alpha_0 = 1$ and the first term is zero. By Lemma 3, there exist numerical constants $C_1, C_2 > 0$ such that

$$\left(\sum_{t=0}^{T-1}\alpha_t^2\prod_{\tau=t+1}^{T-1}(1-\alpha_\tau)\right)^{1/2} \leq C_1\alpha_T^{1/2}, \qquad \left(\sum_{t=0}^{T-1}\gamma_t\prod_{\tau=t+1}^{T-1}(1-\alpha_\tau)\right) \leq C_2\gamma_T\alpha_T^{-1}.$$

Therefore, for all $T \geq 1$, we have

$$\mathbb{E}\left[\|\hat e_T\|\right] \leq C_1\sigma\alpha_T^{1/2} + C_2\ell\gamma_T\alpha_T^{-1}.$$

$$\begin{aligned}
\sum_{t=0}^{T-1}\gamma_t\mathbb{E}\left[\|\hat e_t\|\right] &\leq C_1\sigma\sum_{t=0}^{T-1}\gamma_t\alpha_t^{1/2} + C_2\ell\sum_{t=0}^{T-1}\gamma_t^2\alpha_t^{-1} \\
&\leq C_1\sigma\gamma\sum_{t=0}^{T-1}(t+1)^{-3/4}(t+1)^{-1/4} + C_2\ell\gamma^2\sum_{t=0}^{T-1}(t+1)^{-3/2}(t+1)^{1/2} \\
&\leq \left(C_1\sigma\gamma + C_2\ell\gamma^2\right)\log(T).
\end{aligned}$$

$\square$

**Lemma 3** (Lemma 15 in [19])**.** *Let* $q \in [0,1)$, $p \geq 0$, $\gamma_0 > 0$ *and let* $\eta_t = \left(\frac{2}{t+2}\right)^q$, $\gamma_t = \gamma_0\left(\frac{1}{t+1}\right)^p$ *for every integer* $t$. *Then for any integers* $t$ *and* $T \geq 1$, *it holds*

$$\sum_{t=0}^{T-1}\gamma_t\prod_{\tau=t+1}^{T-1}(1-\eta_\tau) \leq C\gamma_t\eta_T^{-1},$$

*where* $C := 2^{p-q}(1-q)^{-1}t_0\exp\left(2^q(1-q)t_0^{1-q}\right) + 2^{2p+1-q}(1-q)^{-2}$ *and* $t_0 := \max\left\{\left(\frac{p}{(1-q)2^q}\right)^{\frac{1}{1-q}}, 2\left(\frac{p-q}{(1-q)^2}\right)\right\}^{\frac{1}{1-q}}.$

# D Proofs for Scalar AMSGrad and AdaGrad in Section 4

The following is an extended version of Theorem 5 including $\gamma_t = \frac{\gamma}{(t+1)^\alpha}$ with $0 < \alpha < 1$.

**Theorem 7.** *Under Assumption 1, if we run AMSGrad-norm with $\gamma_t = \frac{\gamma}{(t+1)^\alpha}$, $v_0 > 0$ and $\beta_1 = \beta_2 = 0$ in the deterministic setting, then for any $\gamma > 0$ and $0 < \alpha < 1$, if $v_0 < \gamma\ell$*

$$\frac{1}{T}\sum_{t=0}^{T-1}\|\nabla f(x_t)\|^2 \leq \frac{2\Delta}{\gamma T^{1-\alpha}}\max\{v_0, \sqrt{2\ell\Delta}\},$$

*if $v_0 \geq \gamma\ell$*

$$\frac{1}{T}\sum_{t=0}^{T-1}\|\nabla f(x_t)\|^2 \leq \frac{\left(\frac{\ell\gamma}{v_0}\right)^{\frac{1}{\alpha}}\gamma^2\ell^2}{T} + \frac{2(M+\Delta)}{\gamma T^{1-\alpha}}\max\{\gamma\ell, \sqrt{2\ell(M+\Delta)}\},$$

*where*

$$M = \begin{cases} \ell\gamma^2\left(1 + \log\left(\frac{\ell\gamma}{v_0}\right)\right), & \text{when } \alpha = 1/2, \\[2ex] \dfrac{\ell\gamma^2}{2\left(1 - 2^{1-2\alpha}\right)}, & \text{when } 1/2 < \alpha < 1, \\[2ex] \dfrac{\gamma(\ell\gamma)^{\frac{1}{\alpha}-1}}{2(1-2\alpha)v_0^{\frac{1}{\alpha}-2}}, & \text{when } 0 < \alpha < 1/2. \end{cases}$$

**Proof.** The effective stepsize of AMSGrad-norm contains a maximum over all gradient norms in the denominator. As it is desirable to find a lower bound for the effective stepsize, we begin by bounding the gradient norms.

Let $\tau$ be the first iteration where the effective stepsize is less or equal to $1/\ell$, i.e., $\eta_{\tau-1} > 1/\ell$ and $\eta_\tau \leq 1/\ell$. First, we assume $\tau \geq 1$, i.e., $v_0 < \gamma\ell$. The time stamp $\tau$ itself is naturally bounded by

$$\eta_{\tau-1} = \frac{\gamma}{\tau^\alpha v_\tau} > \frac{1}{\ell} \implies \tau < \left(\frac{\ell\gamma}{v_\tau}\right)^{\frac{1}{\alpha}} \leq \left(\frac{\ell\gamma}{v_0}\right)^{\frac{1}{\alpha}}.$$

We have

$$\sum_{t=0}^{\tau-1}\|\nabla f(x_t)\|^2 \leq \tau\gamma^2\ell^2 \leq \left(\frac{\ell\gamma}{v_0}\right)^{\frac{1}{\alpha}}\gamma^2\ell^2. \tag{10}$$

By $\ell$-smoothness of $f(\cdot)$,

$$f(x_{t+1}) \leq f(x_t) + \langle\nabla f(x_t), x_{t+1} - x_t\rangle + \frac{\ell}{2}\|x_{t+1} - x_t\|^2$$

$$= f(x_t) - \eta_t\|\nabla f(x_t)\|^2 + \frac{\ell\eta_t^2}{2}\|\nabla f(x_t)\|^2 \tag{11}$$

$$\leq f(x_t) + \frac{\ell\eta_t^2}{2}\|\nabla f(x_t)\|^2.$$

Therefore,

$$f(x_\tau) - f(x_0) \leq \frac{\ell}{2}\sum_{t=0}^{\tau-1}\eta_t^2\|\nabla f(x_t)\|^2 = \frac{\ell}{2}\sum_{t=0}^{\tau-1}\frac{\gamma_t^2}{v_{t+1}^2}\|\nabla f(x_t)\|^2 \leq \frac{\ell}{2}\sum_{t=0}^{\tau-1}\gamma_t^2$$

$$\leq \begin{cases} \dfrac{\ell\gamma^2}{2}(1+\log\tau), & \text{when } \alpha = 1/2, \\[3mm] \dfrac{\ell\gamma^2}{2\left(1-2^{1-2\alpha}\right)}, & \text{when } 1/2 < \alpha < 1, \\[3mm] \dfrac{\ell\gamma^2\tau^{1-2\alpha}}{2(1-2\alpha)}, & \text{when } 0 < \alpha < 1/2. \end{cases}$$

We denote the right hand side as $M$. Also from (11) and definition of $\tau$, we know that $f(x_t) \leq f(x_\tau)$ for $t \geq \tau$ and therefore, for all $t \geq \tau$,

$$f(x_t) - f^* = f(x_\tau) - f(x_0) + f(x_0) - f^* \leq M + \Delta,$$

which implies

$$\|\nabla f(x_t)\|^2 \leq 2\ell(f(x_t) - f^*) \leq 2\ell(M + \Delta).$$

Therefore, we can bound for all $t \geq 0$,

$$v_t \leq \max\{\gamma\ell, \sqrt{2\ell(M+\Delta)}\}.$$

For $t \geq \tau$, by (11)

$$f(x_{t+1}) - f(x_t) \leq -\frac{\eta_t}{2}\|\nabla f(x_t)\|^2.$$

By telescoping from $t = \tau$ to $T - 1$, we get

$$\begin{aligned} 2\left(f(x_\tau) - f(x_T)\right) &\geq \sum_{t=\tau}^{T-1} \eta_t \|\nabla f(x_t)\|^2 \\ &= \sum_{t=\tau}^{T-1} \frac{\gamma}{(t+1)^\alpha v_{t+1}} \|\nabla f(x_t)\|^2 \\ &\geq \sum_{t=\tau}^{T-1} \frac{\gamma}{T^\alpha v_{t+1}} \|\nabla f(x_t)\|^2 \\ &\geq \sum_{t=\tau}^{T-1} \frac{\gamma}{T^\alpha \max\{\gamma\ell, \sqrt{2\ell(M+\Delta)}\}} \|\nabla f(x_t)\|^2. \end{aligned}$$

Then we have

$$\begin{aligned} \sum_{t=\tau}^{T-1} \|\nabla f(x_t)\|^2 &\leq \frac{2}{\gamma}\left(f(x_\tau) - f(x_T)\right) T^\alpha \max\{\gamma\ell, \sqrt{2\ell(M+\Delta)}\} \\ &\leq \frac{2}{\gamma}\left(f(x_\tau) - f(x^*)\right) T^\alpha \max\{\gamma\ell, \sqrt{2\ell(M+\Delta)}\} \\ &\leq \frac{2(M+\Delta)}{\gamma} T^\alpha \max\{\gamma\ell, \sqrt{2\ell(M+\Delta)}\}. \end{aligned}$$

Combining with (10), we obtain

$$\sum_{t=0}^{T-1} \|\nabla f(x_t)\|^2 \leq \left(\frac{\ell\gamma}{v_0}\right)^{\frac{1}{\alpha}} \gamma^2\ell^2 + \frac{2(M+\Delta)T^\alpha}{\gamma} \max\{\gamma\ell, \sqrt{2\ell(M+\Delta)}\}.$$

When $\tau = 0$, we have

$$2\left(f(x_0) - f(x_T)\right) \geq \sum_{t=0}^{T-1} \frac{\gamma}{T^\alpha v_{t+1}} \|\nabla f(x_t)\|^2 \geq \sum_{t=\tau}^{T-1} \frac{\gamma}{T^\alpha \max\{v_0, \sqrt{2\ell\Delta}\}} \|\nabla f(x_t)\|^2,$$

which implies

$$\sum_{t=0}^{T-1} \|\nabla f(x_t)\|^2 \leq \frac{2\Delta T^\alpha}{\gamma} \max\{v_0, \sqrt{2\ell\Delta}\}.$$

**Remark 3.** *For any $0 < \alpha < 1$, if we compare simplified AMSGrad with $\gamma_t = \frac{\gamma}{(t+1)^\alpha}$ to SGD with $\eta_t = \frac{\eta}{(t+1)^\alpha}$ in the deterministic case (setting $\sigma = 0$ in Theorem 6), we observe that they achieve the same convergence rate. However, the complexity of simplified AMSGrad only includes polynomial term in $\gamma$ and $\ell$, while that of SGD includes an exponential term in $(\eta\ell)^{1/\alpha}$.*

$\square$

In the following, we will first provide the lower bounds for scalar version of AMSGrad (referred to as AMSGrad-norm) with each $\alpha \in (0, 1)$ and discuss why it may fail with $\alpha = 0$ when problem parameters are unknown, which means that it can not achieve the optimal complexity $\mathcal{O}(\epsilon^{-2})$ in the deterministic setting. Second, we show that it also fails to achieve the optimal convergence rate in the stochastic setting when stochastic gradients are unbounded. To make the results more general, we consider the standard scalar AMSGrad with momentum hyper-parameters $\beta_1$ and $\beta_2$, which is presented in Algorithm 2.

Before proceeding to our results, we present a lemma which is handy for conducing lower bounds for SGD-like algorithms with momentum (see Algorithm 4). As long as an upper bound is known for stepsize $\eta_t$, we can derive a lower bound similar to Proposition 1 in [17].

---

**Algorithm 4** General SGD with Momentum

---

1: **Input:** initial point $x_0$, momentum parameters $0 \leq \beta_1 < 1$ and initial moment $m_0$.
2: **for** $t = 0, 1, 2, ...$ **do**
3:     sample $\xi_t$
4:     $m_{t+1} = \beta_1 m_t + (1 - \beta_1)g(x_t; \xi_t)$
5:     obtain stepsize $\eta_t > 0$
6:     $x_{t+1} = x_t - \eta_t m_{t+1}$
7: **end for**

---

**Lemma 4.** *For any $\ell > 0$, $\Delta > 0$ and $T > 1$, there exists a $\ell$-smooth function $f : \mathbb{R} \to \mathbb{R}$, and $x_0$ with $f(x_0) - \inf_x f(x) \leq \Delta$, such that if we run Algorithm 4 with deterministic gradients and $\eta_t \leq \tilde{\eta}_t$ for $t = 0, 1, 2, ..., T - 1$, then we have*

$$\min_{t \in \{0,1,...,T-1\}} |\nabla f(x_t)| \geq \sqrt{\frac{\Delta}{16 \max\{1/\ell, \sum_{t=0}^{T-1} \tilde{\eta}_t\}}}.$$

**Proof.** We construct a quadratic function similar to Proposition 1 in [17]. The following function is considered:

$$f(x) = \frac{x^2}{4 \max\left\{1/\ell, \sum_{t=0}^{T-1} \tilde{\eta}_t\right\}}.$$

Without loss of generality, we assume the initial moment $m_0$ is non-positive, and we set the initial point $x_0$ as

$$x_0 = \sqrt{\Delta \max\left\{1/\ell, \sum_{t=0}^{T-1} \tilde{\eta}_t\right\}}.$$

Otherwise if the initial moment is set to be positive, then we let $x_0$ be negative and follow the same reasoning.

Since $x_0$ is positive, the first gradient direction would be positive, i.e., $\nabla f(x_0) > 0$. Let $\tau$ be the first iteration such that $m_\tau > 0$. By the update rule and definition of $\tau$, it is obvious that $x_t \geq x_0$ for $t \leq \tau - 1$. If $T \leq \tau$, it trivially holds that $\nabla f(x_t) \geq \nabla f(x_0)$ for all $0 \leq t \leq T - 1$. Otherwise,

we have $m_\tau = \beta_1 m_{\tau-1} + (1 - \beta_1)\nabla f(x_{\tau-1}) \le (1 - \beta_1)\nabla f(x_{\tau-1})$. That is to say, the gradient estimation $m_\tau$ used in the $\tau$-th step has the correct direction but its magnitude is no larger than the actual gradient. Starting from the $\tau$-th iteration, $x_t$ will monotonically move left towards the solution. Note that since our stepsize is small enough, i.e.,

$$\eta_t \le \tilde{\eta}_t < 2\max\left\{1/\ell, \sum_{t=0}^{T-1}\tilde{\eta}_t\right\},$$

the updates will remain positive, i.e., $x_t > 0$ for $t \ge \tau$. By the update rule, we note that $x_{t+1} \le x_t$ for $t \ge \tau$, and therefore $\nabla f(x_{t+1}) < \nabla f(x_t)$. We can conclude that for any $t \ge \tau$, we have $m_t \le \nabla f(x_{\tau-1})$. Then for $t \ge \tau - 1$ we have

$$
\begin{aligned}
x_t &= x_{\tau-1} - \sum_{k=\tau-1}^{t-1} \eta_t m_{t+1} \\
&\ge x_{\tau-1} - \sum_{k=\tau-1}^{t-1} \tilde{\eta}_t \nabla f(x_{\tau-1}) \\
&= x_{\tau-1} - \sum_{k=\tau-1}^{t-1} \frac{\tilde{\eta}_t}{2\max\left\{1/\ell, \sum_{t=0}^{T-1}\tilde{\eta}_t\right\}} x_{\tau-1} \\
&\ge \frac{1}{2}x_{\tau-1} \\
&\ge \frac{1}{2}x_0.
\end{aligned}
$$

Then we conclude by

$$|\nabla f(x_t)| = \frac{x_t}{2\max\left\{1/\ell, \sum_{t=0}^{T-1}\tilde{\eta}_t\right\}} \ge \frac{x_0}{4\max\left\{1/\ell, \sum_{t=0}^{T-1}\tilde{\eta}_t\right\}} = \sqrt{\frac{\Delta}{16\max\{1/\ell, \sum_{t=0}^{T-1}\tilde{\eta}_t\}}}.$$

$\square$

Now we proceed to provide the lower bound for deterministic case.

**Theorem 8.** *For any $\ell > 0$, $\Delta > 0$ and $T > 1$, there exists a $\ell$-smooth function $f : \mathbb{R} \to \mathbb{R}$ and $x_0$ with $f(x_0) - \inf_x f(x) \le \Delta$, such that if we run Algorithm 2 with deterministic gradients, $0 < v_0 \le \frac{\ell\gamma}{2}$, and $\gamma_t = \frac{\gamma}{(t+1)^\alpha}$ with $\gamma \le \frac{4\Delta}{v_0}$, we have (1) if $0 < \alpha < 1$, for any $0 \le \beta_1 < 1$ and $0 \le \beta_2 \le 1$, we have*

$$\min_{t\in\{0,1,\dots,T-1\}} |\nabla f(x_t)| \ge \sqrt{\frac{\Delta}{16\max\{1/\ell, \frac{\gamma}{(1-\alpha)v_0}T^{1-\alpha}\}}},$$

*and (2) if $\alpha = 0$, for $\beta_1 = 0$ and any $0 \le \beta_2 \le 1$, we have*

$$\min_{t\in\{0,1,\dots,T-1\}} |\nabla f(x_t)| \ge v_0.$$

**Remark 4.** *From the theorem, we can conclude that the optimal convergence rate $\frac{1}{\sqrt{T}}$ for $\|\nabla f(x_t)\|$ is infeasible for AMSGrad with polynomially decreasing stepsize. When $\alpha = 0$, a similar result can be obtained for the case $\beta_1 \ge 0$, $\beta_2 = 0$ and small enough $v_0$.*

**Proof.** For $\alpha > 0$, we have

$$\eta_t = \frac{\gamma}{(t+1)^\alpha\sqrt{\hat{v}_{t+1}^2}} \le \frac{\gamma}{(t+1)^\alpha v_0}.$$

Let $\tilde{\eta}_t = \frac{\gamma}{(t+1)^\alpha v_0}$ and then we have

$$\sum_{t=0}^{T-1} \tilde{\eta}_t = \sum_{t=0}^{T-1} \frac{\gamma}{(t+1)^\alpha v_0} \le \frac{\gamma}{(1-\alpha)v_0} T^{1-\alpha}.$$

Applying Lemma 4 directly gives us the desired result.

For $\alpha = 0$, we consider function

$$f(x) = \frac{v_0}{\gamma} x^2.$$

Note that since $v_0 \le \frac{\ell\gamma}{2}$, the function is $\ell$-smooth. Let

$$x_0 = \frac{\gamma}{2},$$

which satisfies the condition that $f(x_0) \le \Delta$. Then after one update

$$v_1^2 = \beta_2 v_0^2 + (1-\beta_2)\|\nabla f(x_0)\|^2 = v_0^2$$
$$x_1 = x_0 - \frac{\gamma}{\sqrt{v_1^2}} \nabla f(x_0) = -\frac{\gamma}{2} = -x_0.$$

If we continue this calculation, we find that the iterates will oscillate between $\frac{\gamma}{2}$ and $-\frac{\gamma}{2}$ forever, which finishes the proof. $\qquad\square$

**Proof for Theorem 4**

**Proof.** We consider a two-dimensional function $f : \mathbb{R}^2 \to \mathbb{R}^2$, for $x = (x^1, x^2)^\top \in \mathbb{R}^2$,

$$f(x) = F(x^1),$$

where its function value only depends on the first dimension and we will define $F : \mathbb{R} \to \mathbb{R}$ later. The gradient at $x$ is $\nabla f(x) = (\nabla F(x^1), 0)^\top$. We add the noise only to the second dimension, i.e., $g(x; \xi) = (\nabla F(x^1), \xi)$. For any $t \ge 0$, the probability density function of the noise as

$$p_{\xi_t}(x) = \begin{cases} \frac{1}{s\zeta}\left(\frac{x}{s}\right)^{-1-\frac{2}{\zeta}} e^{-\left(\frac{x}{s}\right)^{-\frac{2}{\zeta}}}, & x \ge 0; \\ \frac{1}{s\zeta}\left(\frac{-x}{s}\right)^{-1-\frac{2}{\zeta}} e^{-\left(\frac{-x}{s}\right)^{-\frac{2}{\zeta}}}, & x < 0, \end{cases}$$

where $s = \frac{\sigma}{\sqrt{\Gamma\left(1-\frac{\zeta}{2}\right)}}$. Note that the distribution is symmetric and $\mathbb{E}\left[\xi_t\right] = 0$. Also, we note that $|\xi_t|$ follows the Fréchet distribution [15] with cumulative distribution function

$$\Pr(|\xi_t| \le x) = e^{-\left(\frac{x}{s}\right)^{-\frac{2}{\zeta}}},$$

and

$$\begin{aligned} \mathrm{Var}\left[\xi_t\right] &= \mathbb{E}\left[|\xi_t|^2\right] - \left(\mathbb{E}\left[\xi_t\right]\right)^2 \\ &= s^2\Gamma\left(1-\frac{\zeta}{2}\right) - \left(\mathbb{E}\left[\xi_t\right]\right)^2 \\ &\le s^2\Gamma\left(1-\frac{\zeta}{2}\right) \\ &\le \sigma^2, \end{aligned}$$

where we used the exact second moment for Fréchet distribution.

Next, we will show that $\tilde{\xi}_t := \max_{0 \leq k \leq t}\{|\xi_k|\} \geq \Omega\left(\frac{1}{(t+1)^{\varsigma - 1/2}}\right)$ with probability $\frac{1}{2}$. We know that $\tilde{\xi}_t$ also follows Fréchet distribution with CDF

$$\Pr(\tilde{\xi}_t \leq x) = \exp\left(-\left(\frac{x}{s \cdot (t+1)^{\frac{\varsigma}{2}}}\right)^{-\frac{2}{\varsigma}}\right).$$

Then for constant $C > 0$,

$$
\begin{aligned}
\Pr(\tilde{\xi}_t \leq C \cdot (t+1)^{\varsigma - \frac{1}{2}}) &= \exp\left(-\left(\frac{C \cdot (t+1)^{\varsigma - \frac{1}{2}}}{s \cdot (t+1)^{\frac{\varsigma}{2}}}\right)^{-\frac{2}{\varsigma}}\right) \\
&= \exp\left(-\left(\frac{C}{s}\right)^{\frac{2}{\varsigma}} (t+1)^{\frac{1}{\varsigma} - 1}\right) \\
&\leq \frac{1}{4(t+1)^2},
\end{aligned}
$$

where the last inequality is by selecting $C = \frac{s\left(e\left(\frac{1}{\varsigma} - 1\right)\right)^{\frac{\varsigma}{2}}}{\sqrt{2}}$ and using $\exp\left(-\frac{2^{m+1}}{em} \cdot t^m\right) \leq \frac{1}{4t^2}$ for any $t > 0$ and $0 < m < 1$. Then using union bound, we have

$$\Pr(\tilde{\xi}_t > C \cdot (t+1)^{\varsigma - \frac{1}{2}} \quad \text{for} \quad 0 \leq t \leq T - 1) \geq 1 - \sum_{t=0}^{T-1} \frac{1}{4(t+1)^2} \geq \frac{1}{2}.$$

Now we have shown that with some probability, the noise is large enough. We can use this property to provide an upper bound $\tilde{\eta}_t$ for the stepsize as follow

$$
\begin{aligned}
\eta_t &= \frac{\gamma}{\sqrt{t+1}\sqrt{\tilde{v}_{t+1}}} \\
&= \frac{\gamma}{\sqrt{t+1}\sqrt{\max_{0 \leq k \leq t}\{\beta_2 v_k + (1-\beta_2)\|g(x_k;\xi_k)\|^2\}}} \\
&\leq \frac{\gamma}{\sqrt{t+1}\sqrt{\max_{0 \leq k \leq t}\{(1-\beta_2)\|g(x_k;\xi_k)\|^2\}}} \\
&\leq \frac{\gamma}{\sqrt{t+1}\sqrt{\max_{0 \leq k \leq t}\{(1-\beta_2)\|\xi_k\|^2\}}} \\
&= \frac{\gamma}{\sqrt{t+1}\sqrt{(1-\beta_2)}\tilde{\xi}_t} \\
&\leq \frac{\gamma}{C(t+1)^\varsigma\sqrt{(1-\beta_2)}} \triangleq \tilde{\eta}_t,
\end{aligned}
$$

This implies

$$\sum_{t=0}^{T-1} \tilde{\eta}_t \leq \frac{\gamma}{(1-\varsigma)C\sqrt{(1-\beta_2)}}\left(T^{1-\varsigma} - \varsigma\right).$$

We observe that the update with AMSGrad-norm in function $f$ corresponds to applying general SGD with momentum (Algorithm 4) to function $F$ with stepsize $\eta_t$. Therefore, we can pick a hard instance $F$ according to Lemma 4, and by noting that $\|\nabla f(x)\| = |\nabla F(x^1)|$ we reach our conclusion.

$\square$

**Remark 5.** *As we see above, the function $F$ in the proof is constructed by Lemma 4. We note that even assuming the gradients of $f$ to be bounded, i.e., $\|\nabla f(x)\| \leq K$ for all $x$, will not prevent the slow convergence in Theorem 4. This is because in the proof of Lemma 4 all iterates stay between $[0, x_{\tau-1}]$ (e.g., $\tau = 1$ if $m_0 = 0$), so we can construct any Lipschitz function outside of this segment.*

**Algorithm 5** AdaGrad-norm

---

1: **Input:** initial point $x_0$, $v_0 > 0$ and $\eta > 0$
2: **for** $t = 0, 1, 2, ...$ **do**
3:     sample $\xi_t$
4:     $v_{t+1}^2 = v_t^2 + \|g(x_k; \xi_k)\|^2$
5:     $x_{t+1} = x_t - \frac{\eta}{\sqrt{v_{t+1}^2}} g(x_t; \xi_t)$
6: **end for**

---

**Proof for Proposition 3**

**Proof.** Define a function $\widetilde{f} : \mathbb{R}^d \times \mathbb{R} \to \mathbb{R}$ such that $\widetilde{f}(x, y) = f(x) - \frac{\ell}{2} y^2$. Since the $\widetilde{f}$ is $\ell$-smooth and $\ell$-strongly concave about $y$, the condition number is defined to be $\kappa = 1$. Applying AdaGrad-norm to $f$ is equivalent to applying NeAda-AdaGrad (Algorithm 3 in [68]) to $\widetilde{f}$ with $y_t \equiv 0$. For every $x$, we know $y^*(x) \triangleq \arg\max_y \widetilde{f}(x, y) = 0$. Then $\mathcal{E} \triangleq \sum_{t=0}^{T-1} \frac{\ell^2 \|y_t - y^*(x_t)\|^2}{2v_0} = 0$. Plugging in $\kappa = 1$, $\mathcal{E} = 0$ and batchsize $M = 1$ to Theorem 3.1 in [68], we reach the conclusion.

$\square$

