# OpenReview forum: "Two Sides of One Coin: the Limits of Untuned SGD and the Power of Adaptive Methods"
_NeurIPS.cc/2023/Conference — NeurIPS 2023 poster_

### Official Review · Reviewer_hiCA · 2023-06-29

**Soundness:** 3 good
**Presentation:** 3 good
**Contribution:** 2 fair
**Rating:** 5
**Confidence:** 4

**Summary:**

This paper shows mainly two things:
(a) SGD suffers from an exponential dependence on the initial stepsize if it is not tuned to be smaller than the learning rate. This exponential dependence is unavoidable.
(b) Methods with gradient normalization and running gradient sum normalization, such as Normalized SGD, AMSGrad, and AdaGrad, suffer no such exponential dependence on the smoothness constant.

The paper also presents a novel analysis of AMSGrad that removes the bounded gradients assumption.

**Strengths:**

1. The result on AMSGrad is new and a welcome addition to the literature.
2. The paper's emphasis on the benefits of normalization even in the deterministic setting is good, since this is a point quite overlooked in the community.

**Weaknesses:**

1. The result on the exponential dependence on the smoothness constant is a known consequence of another result in the literature. Under the assumption on the stochastic gradients $\mathbb{E} \|g(x)\|^2 <= 2 A (f(x)-f_*) + B \|\nabla f(x\||^2 + C$. Note that bounded stochastic gradient variance corresponds to $A=L$, $B=0$ and $C=\sigma$ (since bounded variance implies $E||g(x)||^2 <= ||\nabla f(x)||^2 + C <= 2 L (f(x)-f_*) + C$). The result of Theorem 2 in [1] gives for this choice ($A=L$, $B=0$, $C=\sigma$) a rate of $\frac{(1+\gamma^2 L^2)^K}{\gamma K} \delta_0 + L \gamma C$ where $\delta_0$ is the initial suboptimality. The lower bound is also known, see [2, Theorem 5].

[1] Khaled and Richtárik. Better Theory for SGD in the Nonconvex World. arXiv:2002.03329
[2] Vaswani, Benjamin Dubois-Taine, and Babanezhad. Towards Noise-adaptive, Problem-adaptive (Accelerated) Stochastic Gradient Descent. arXiv:2110.11442.



**Questions:**

1. Please address the difference between your results and the results I've mentioned in the weaknesses section.

---

> ### Author Rebuttal · Authors · 2023-08-09
>
> Thanks for the comments.
>
> > **The result on the exponential dependence on the smoothness constant is a known consequence of another result in the literature...**
>
>
> We thank the reviewer for pointing out the two relevant references and we will add more discussions about them in the revision. However, we are afraid that the reviewer might have overlooked some fundamental differences between our results and theirs.
>
> **In reference [1]**, Theorem 2  states that: with a constant stepsize $\eta$, an upper bound of $\mathcal{O}\left(\eta + \frac{(1+\ell^2\eta^2)^T}{\eta T} \right)$ holds after $T$ iterations. There are key differences from our Theorem 1:
>
> 1. They consider constant stepsize $\eta$, while we consider polynomially decreasing stepsize $\eta/\sqrt{t}$.
>
> 2. The exponential terms are very different. Their result contains the term $(1+\ell^2\eta^2)^T$ with the exponent $T$, while ours is $(4e)^{2\eta^2\ell^2}$ with exponent $2\eta^2\ell^2$.
>
> 3. When $\eta$ is relatively large, their result diverges, whereas ours consistently converges.
>
>
> Note that we have also discussed in Remark 1 Line 183 - 185: "We do not
>  consider constant stepsize, i.e., $\alpha = 0$, because it is well known to diverge even in the deterministic setting if the stepsize is agnostic to the problem parameter [1, 51].", highlighting the divergent behavior of the constant stepsize. In contrast, the diminishing stepsizes we consider are more interesting, and always lead to convergence despite the presence of the exponential constant.
>
>
> **In reference [2]**, Theorem 5 states that there exists a quadratic function $f(x)$ such that, fixing the total number of iterations $T$, running gradient descent with stepsize $\eta\_t = \frac{\nu}{\ell}\left(\frac{\beta}{T} \right)^{t/T}$, with $\nu$ and $\beta$ being some constants, will satisfy $\\\|x\_{\tilde t + 1} - x^\*\\\| \geq 2^{\tilde{t}}\\\|x\_0 - x^*\\\|$ at an iteration $\tilde{t} = \Theta(T/\ln(T))$. The primary distinctions from our lower bound are:
>
> 1. The settings are different. They consider strongly convex setting with exponentially decreasing stepsizes that requires a prefixed $T$, but we consider nonconvex setting with polynomially decreasing stepsizes.
>
> 2. Their result features the exponential term $2^{\tilde{t}}$ at a specific iteration $\tilde{t}$, but their upper bound (as per Theorem 4 in [2]) actually does not include any exponential term at the last iteration $T$. This implies that after $\tilde t$ iteration, gradient descent can  converge very quickly in their setting. In contrast, both our upper and lower bounds include an exponential term multiplied by $T^{-1/4}$ at the last iteration.
>
>
> We hope our clarification on these differences addresses the reviewer's concern and we are happy to discuss more if the reviewer has further questions.

---

> > ### Comment · Reviewer_hiCA · 2023-08-10
> >
> > Thank you for your response.
> >
> > 1. It is trivial to just instead use the constant stepsize $\frac{\eta}{\sqrt{T}}$ when the horizon $T$ is known, and obtain a very similar exponent to the one you have. Observe that $1+x \leq e^x$ and therefore $(1+ \ell^2 \frac{\eta^2}{T})^{T} \leq e^{\ell^2 \eta^2}$. I don't think the analysis with decreasing versus constant stepsizes with known time horizon is different to qualify as its own paper.
> > 2. A strongly convex function is in a smaller function class than nonconvex functions, i.e. a lower bound that constructs a strongly convex function might be too tight, but is never too loose. Therefore, if it shows divergence in case the stepsize is misspecified, this holds for nonconvex objectives. And the main message of their result as applied to your setting would be the limit of stepsize misspecification by adaptivity, not the exact convergence rate.

---

> > > ### Author Response · Authors · 2023-08-11
> > >
> > > Thanks for actively engaging in the discussion.
> > >
> > >
> > >
> > > > **It is trivial to just instead use the constant stepsize $\frac{\eta}{\sqrt{T}}$ when the horizon  is known, and obtain a very similar exponent to the one you have. Observe that $1+x \leq e^x$ and therefore $(1+\ell^2 \frac{\eta^2}{T})^T \leq e^{\ell^2 \eta^2}$. I don't think the analysis with decreasing versus constant stepsizes with known time horizon is different to qualify as its own paper.**
> > >
> > >
> > > Regarding the upper bound in [1], we agree that the exponential term looks similar provided that we know $T$ and can pick the constant stepsize $\eta/\sqrt{T}$. At the same time, we humbly believe that extending the analysis in [1] to the case of diminishing stepsize $\eta_t = \eta/\sqrt{t}$ is not completely straightforward. However, we would like to highlight that [1] only provides the upper bound, while in order to make a strict separation between untuned SGD and adaptive methods, it is necessary to establish the lower bound for untuned SGD. This is because the upper bound might be loose and alone **does not imply the exponential dependence on $\ell$ multiplied with $\epsilon^{-4}$ is tight**. This lower bound construction is the key argument of our work to showcase the fundamental difference between untuned SGD and adaptive methods. To our knowledge, our **lower bound construction is novel and distinct** from those in the literature and applies to the fundamental non-convex smooth setting (with classical polynomially diminishing stepsizes). Importantly, our upper and lower bounds  match, providing a comprehensive analysis of nonconvex untuned SGD. We believe that this contribution gives conclusive evidence about the limits of parameter agnostic methods under this setting, a topic that was not sufficiently discussed in the optimization literature.
> > >
> > >
> > > > **A strongly convex function is in a smaller function class than nonconvex functions, i.e. a lower bound that constructs a strongly convex function might be too tight, but is never too loose. Therefore, if it shows divergence in case the stepsize is misspecified, this holds for nonconvex objectives. And the main message of their result as applied to your setting would be the limit of stepsize misspecification by adaptivity, not the exact convergence rate.**
> > >
> > >
> > > It is worth noting that the upper bound in [2] (Theorem 4 in the paper) does not include the exponential term in the last iterate $T$, which implies there is also no exponential term in the lower bound for the last iterate. Their lower bound's exponential term only emerges before an iterate $\tilde{t} < T$. If we adopt their lower bound case $f(x) = \frac{\ell}{2}(x-a)^2$ with our stepsize $\eta/\sqrt{t}$, it will lead to a lower bound of $\Omega\left((2\eta \ell)^{-2} \log^2\frac{\exp(\ell^2 \eta^2)}{\epsilon}\right)$, in which the exponential term will be **forgotten exponentially fast** and there exists a huge gap with the upper bound of $\mathcal{O} (\exp(\ell^2)\epsilon^{-1/4})$. The multiplication between the exponential term and $\epsilon^{-1/4}$ is especially important because  $\ell$ is usually large and the target accuracy is small. To achieve a matching term in the lower bound is challenging and we carefully construct our lower bound with a **nonconvex** example.

---

> > > > ### Comment · Reviewer_hiCA · 2023-08-14
> > > >
> > > > Okay, please cite both the upper bound of [1] and the lower bound of [2] clearly and compare them with your results. Please contextualize your contributions exactly in the main paper as you have done here. As my main concern is addressed, I will raise my score.

---

### Official Review · Reviewer_vyoV · 2023-07-06

**Soundness:** 3 good
**Presentation:** 3 good
**Contribution:** 3 good
**Rating:** 6
**Confidence:** 3

**Summary:**

The authors investigate the behavior of untuned SGD in the smooth nonconvex setting and show a new result on the convergence rate of SGD w.r.t. to gradient norm, where there is an exponential dependence on the smoothness constant. They further argue that the exponential dependence is unavoidable through a constructed class of 1-dimensional nonconvex functions. The paper then examines NSGD, AMSGrad and AdaGrad and shows that the exponential dependence can be avoided through adaptiveness, albeit without any information about the problem parameters.

**Strengths:**

This paper offers an interesting theoretical perspective on the explosive gradient problem and the nonconvergence properties of SGD. The authors complement their theoretical results and ideas with numerical illustrations, and the paper is easy to follow. Their results seem well-justified, but I did not check their proofs in the appendix. I believe this work is of interest to the NeurIPS community.

**Weaknesses:**

1) The current state of numerical experiments seem preliminarily and is only done on one dataset MNIST on a small-network. I would like to see a more comprehensive investigation into larger practical networks, perhaps from [6, 24, 54].

**Questions:**

1) In figure 1, what is the size of each layer in the 3-layer neural network? What effect does over-parameterization have on the untuned SGD behavior?

**Limitations:**

Yes, the authors have addressed their limitations through the checklist.

---

> ### Author Rebuttal · Authors · 2023-08-09
>
> Thanks for the recognition of our work.
>
> > **The current state of numerical experiments seem preliminarily and is only done on one dataset MNIST on a small-network. I would like to see a more comprehensive investigation into larger practical networks, perhaps from [6, 24, 54].**
>
>
> To complement our experiments on MNIST, we used the CIFAR-10 dataset (Krizhevsky et al., 2009), as suggested in [24], to train a 50-layer
> ResNet (He et al., 2016), which is more common and large-scale than models in [6, 54]. In these experiments, we observed a similar exponential explosion phenomenon with SGD when using large stepsizes. In contrast, adaptive methods demonstrated robustness to changes in stepsizes. The detailed experimental results can be found in the PDF of our general response to all reviewers.
>
> > **In figure 1, what is the size of each layer in the 3-layer neural network? What effect does over-parameterization have on the untuned SGD behavior?**
>
> In Figure 1 of our paper, the 3-layer neural network is structured with layer sizes as follows: 784 (input size) $\rightarrow$ 512 $\rightarrow$ 256 $\rightarrow$ 10.
> In our new experiment, when we use a 50-layer ResNet -- an over-parameterized neural network with more than 23 million parameters -- we observed behaviors consistent with our findings from smaller networks.
>
> **References**
>
> - Krizhevsky, Alex, et al. "Learning multiple layers of features from tiny images. 2009.
> - He, Kaiming, et al. "Deep residual learning for image recognition." CVPR. 2016.

---

> > ### Comment · Reviewer_vyoV · 2023-08-20
> > **Response to Authors' Rebuttal**
> >
> > I acknowedge the response by the authors and have considered other reviewers' comments. I intend to keep my original evaluation.

---

### Official Review · Reviewer_WWqH · 2023-07-06

**Soundness:** 4 excellent
**Presentation:** 4 excellent
**Contribution:** 4 excellent
**Rating:** 7
**Confidence:** 2

**Summary:**

This paper analyzes the complexity of finding an $\epsilon$-stationary point for untuned SGD and compares that with three families of adaptive methods - NSGD, AMSGrad and AdaGrad. Compared to previous convergence analysis results for tuned SGD and Adaptive methods: this work gets rid of several assumptions that hides the true convergence behaviour of these algorithms. Specifically, the authors do not assume the step-size for SGD to be dependent on the smoothness parameter (hence untuned) and do not assume bounded gradients for the adaptive methods. These leads to an interesting comparison for convergence of these algorithms.

**Strengths:**

1) The authors show that untuned SGD converges to an $\epsilon$ stationary point in $O(e^{\eta^2 l^2}\epsilon^{-4})$ iterations. Although this algorithm does have optimal dependece on $\epsilon$ , it has a disastrous exponential term wrt the smoothness parameter $\eta^2 l^2$, Hence, the assumption on bounded gradients or chosing the $\eta$ to depend on $l$ is problematic, because we may not have prior knowledge of $l$. They show that even for a smooth 1D-function, the assumption of bounded gradient is problematic. This is indeed true and the experiment in figure-1 supports this claim.

2) Adaptive gradient methods adjust their step-size based on observed gradients and hence can decrease the stepe-size when encountered with a large stepe-size preventing blow-up. This work does not assume bounded gradeint assumption for these methods and show that the convergence rate does not exponenetially depend on the smoothness parameter making it more stable than untuned SGD.

The core strength of the paper lies in removing assumption on bounded gradients revealing true dependency of these algorithms on smoothness parameter $l$ which highlights the advantage of adaptive methods over untuned SGD.

**Weaknesses:**

Disclaimer: I am unfamilair with recent developments in this specific diretion. But still i really enjoyed reading this work and I feel it improves over existence convergence results. I have a few questions that i encountered but I don't list them as major weaknesses:

1) the constructed function $f(x) $ in Figure-2 does not look so. Is there an extension of the function outside the segment-1. If so, the author should mention that. Although the equation for sregment-4 and segment-1 are still provided, segment-2 and 3 are still missing.

2) From theorem-1, the main reason untuned SGD may blow up is that some initial $\eta \geq \frac{1}{l}$. But in practice SGD also converges with a small constant learning rate which does fall into this regime and in this regime it does not blow up. However, it is mostly observed that initial large learning rate SGD performs better in terms of generalization [1] . If practitioneers use a small enough (but contant) leanring rate $\eta \leq \frac{1}{l}$, then the whole issue of gradient blow-up can be avoided. Even in such practical cases, no prior knowledge of $l$ is required to select step-size $\eta$. So is it true that achieving generalization is the bottleneck in chosing step-size as large as possible ?



[1] Li, Yuanzhi, Colin Wei, and Tengyu Ma. "Towards explaining the regularization effect of initial large learning rate in training neural networks." Advances in Neural Information Processing Systems 32 (2019).


**Questions:**

See the points above for the questions.

**Limitations:**

i believe the whole problem of gradient blow-up can be avoided by just using small enough practical $\eta$ according to Thoerem-1. But that would hurt generalization in overparameterized networks. Hence, I believe comparison of constant step SGD $\eta$ and other adaptive gradient methods should also be done in terms of generalization and not only convergence. This would give us the full picture.

---

> ### Author Rebuttal · Authors · 2023-08-09
>
>
> Thanks for the recognition of our work.
>
> > **the constructed function $f(x)$ in Figure-2 does not look so. Is there an extension of the function outside the segment-1. If so, the author should mention that. Although the equation for sregment-4 and segment-1 are still provided, segment-2 and 3 are still missing.**
>
> We thank the reviewer for mentioning this, and we will put the formal definitions for Segments-2 and 3 in the main body of the paper. In the current version, their definitions can be found in the proof for Theorem 2 (in Appendix B.2 of the supplementary material). They are constructed to connect Segment-1 to Segment-4 and guarantee the overall function is continuous and $\ell$-smooth.
>
>
> > **From theorem-1, the main reason untuned SGD may blow up is that some initial $\eta \geq \frac{1}{l}$. But in practice SGD also converges with a small constant learning rate which does fall into this regime and in this regime it does not blow up. However, it is mostly observed that initial large learning rate SGD performs better in terms of generalization [1] . If practitioneers use a small enough (but contant) leanring rate $\eta \geq \frac{1}{l}$, then the whole issue of gradient blow-up can be avoided. Even in such practical cases, no prior knowledge of $l$ is required to select step-size $\eta$. So is it true that achieving generalization is the bottleneck in chosing step-size $\eta$ as large as possible?**
>
> If we interpret the reviewer's question accurately, in the last sentence, the reviewer is asking "is it true that achieving generalization is the bottleneck in chosing step-size $\eta$ as _small_ as possible?".
> From the optimization perspective, a reasonable stepsize ($\eta \leq 1/\ell$) avoids blow-up, but an excessively small stepsize  results in slower convergence. This is also indicated in Theorem 1 that when $\eta \leq 1/\ell$, the bound includes an $\eta^{-1}$ factor, signaling a slowdown. The slow convergence is intuitively expected in practical scenarios when an extremely small stepsize is used. Hence, achieving generalization is not the only hindrance; a proper stepsize, e.g. $\Theta(1/\ell)$, is required to achieve fast optimization.
>
> When taking generalization into account, the situation becomes more complicated and exceeds the scope of this paper. However, we agree that the relationship between stepsize and generalization is an intriguing research topic that has been actively and extensively examined (Jastrzębski et al., 2017; He et al., 2019; Li et al., 2019; Nakkiran, 2020).
> It would an interesting future direction to balance optimization and generalization bounds in different stepsize regimes.
>
> **References**
>
> - Jastrzębski, Stanisław, et al. "Three factors influencing minima in sgd." arXiv preprint arXiv:1711.04623. 2017.
> - He, Fengxiang, et al. "Control batch size and learning rate to generalize well: Theoretical and empirical evidence." NeurIPS. 2019.
> - Li, Yuanzhi, et al. "Towards explaining the regularization effect of initial large learning rate in training neural networks." NeurIPS. 2019.
> - Nakkiran, Preetum. "Learning rate annealing can provably help generalization, even for convex problems." arXiv preprint arXiv:2005.07360. 2020.

---

> > ### Comment · Reviewer_WWqH · 2023-08-16
> >
> > I acknowledge that I read the author's rebuttal and based on other reviewer's comments, I intend to keep my current score.

---

### Official Review · Reviewer_viw9 · 2023-07-07

**Soundness:** 4 excellent
**Presentation:** 4 excellent
**Contribution:** 4 excellent
**Rating:** 7
**Confidence:** 4

**Summary:**

This work analyses the rate of SGD and other adaptive SGD methods in reducing $\\mathbb{E}\\|\\nabla f(x_t)\\|$, where $f$ is non-convex $L$-smooth, and shows that SGD has an exponential dependence on $L$ when the stepsize is not properly tuned, while other adaptive methods do not incur this exponential dependence.

**Strengths:**

The analysis presented in this work is fresh and interesting. The authors make a compelling case that removing the bounded gradient assumption is perhaps essential in revealing a crucial advantage of adaptive SGD methods over standard SGD.

As I explain in the weakness section, I think the qualitative idea behind the described phenomenon is known. However, the quantitative results are, as far as I know, new and, in my opinion, very interesting. I think the qualitative understanding gained by the quantitative analysis is valuable, so I vote for the paper to be accepted.

**Weaknesses:**

At a high level, one can argue that the result was "known" in the following sense.

When a LR schedule of $\\eta_k=\\eta/\\sqrt{k}$ is used, $\\eta_k$ will be large for small $k$. It is known that SGD exhibits divergent behavior when the stepsize exceeds $2/L$, so SGD will initially diverge. The worst-case amount of divergence should be exponential in $L$. Once the $\\eta_k$ diminishes to a level below the divergent threshold, then the algorithm needs to recover from the initial exponential divergence, hence the rate.

On the other hand, adaptive methods should not exhibit such divergent behavior (at least not to an exponential extent). Therefore, there is no initial exponential divergence, so there is no catch-up to play at the later stage of the algorithm.

I think it would be worthwhile for the authors to discuss the view that the exponential constant corresponds to the amount of catch-up SGD needs to make.


**Questions:**

Is the line of reasoning the authors lay out specific to non-convex SGD? It seems to me that a similar type of exponential dependence could be shown for the smooth-convex setup.

**Limitations:**

.

---

> ### Author Rebuttal · Authors · 2023-08-09
>
> Thanks for the recognition of our work.
>
> > **At a high level, one can argue that the result was "known" in the following sense... I think it would be worthwhile for the authors to discuss the view that the exponential constant corresponds to the amount of catch-up SGD needs to make.**
>
> We agree that this is an intuition behind our results. We note that SGD with a constant stepsize may diverge exponentially, but it is less apparent that an exponential term persists even with a _diminishing_ stepsize, while simultaneously achieving an order-optimal convergence.
> Furthermore, it is noteworthy that the exponential term is _multiplied_ by the optimal rate in both our upper and lower bounds,
> which is nontrivial.
>
> In general, our results reveal that a strict advantage of adaptive methods is removing the exponential factor, not achieving an order-optimal rate parameter-agnostically as commonly perceived  -- an insight that might seem intuitive in hindsight but to our knowledge has not been formally proved in the existing literature -- and we believe it is a valuable one that deserves recognition.
>
> > **Is the line of reasoning the authors lay out specific to non-convex SGD? It seems to me that a similar type of exponential dependence could be shown for the smooth-convex setup.**
>
>
> We appreciate the reviewer's insightful question. Our focus in this paper is on the nonconvex setting  because most of the motivating examples are in the nonconvex regime.
>
> After some preliminary study, we believe that the upper bound analysis can be extended to the smooth convex setting. Analogous to the proof of Theorem 1, the analysis can be divided into two stages: the first stage is when the stepsize is still larger than $1/\ell$; the second stage is when the stepsize has decreased to be small enough. Let $\tau$ be the first iteration when the stepsize is less than $1/\ell$. In the first stage, by employing techniques similar to those in Theorem 1 and leveraging convexity, we can bound  $\\\|x\_{\tau} - x^\*\\\|^2$ by an exponential term.
> The term  $\\\|x\_\tau - x^\*\\\|$ then serves as an initial distance for the second stage, and we can analyze the convergence in this stage with
> the classic analysis in the stochastic convex setting, leading to a complexity of $\mathcal{O}(1/\epsilon^2)$ for finding an epsilon optimal point (i.e., $f(x) - f^\* \leq \epsilon$). Combining two stages together  leads to a total complexity of $\epsilon^{-2}$ multiplied by an exponential term.
>
>
> Our original nonconvex hard instance for the lower bound does not apply here. However, we may construct a simple instance in the convex setting: a two-dimensional additive example such as $\ell x\_1^2 + \lambda x\_2^2$. This instance results in a lower bound of the order $\exp{(\ell^2)}\log^2\epsilon^{-1} + \epsilon^{-4}$ for finding an $\epsilon$-stationary point or $\exp{(\ell^2)}\log\epsilon^{-1} + \epsilon^{-2}$ for finding an $\epsilon$-optimal point. We note that the highest-order term in $\epsilon$ is not multiplied by the exponential term. Therefore, we suspect there is a small gap with this simple hard instance, making it an interesting future direction to explore.

---

### Official Review · Reviewer_CaFA · 2023-07-09

**Soundness:** 3 good
**Presentation:** 2 fair
**Contribution:** 2 fair
**Rating:** 4
**Confidence:** 3

**Summary:**

The article under review presents results to show that untuned SGD may be less adapted than normalized versions of it when solving smooth non-convex problems.
In order to prove this, the authors present several results, from upper to lower bounds on smooth non-convex problems to find a critical point, without the knowledge of the smoothness parameter. They show that while SGD may suffer from the non-adaptivity of the step-size, its normalized version get rid of this problem.

**Strengths:**

- The paper's description of the problem it wants to tackle is good, and the questions addressed are well introduced.
- While I find it a weakness too (see below), it is remarkable that the authors present both lower bounds and upper bounds on a lot of different settings.
- The figures summarize well the main idea of the negative result on untuned SGD and the principle of the lower bound
- The table helps to navigate in this hairy paper


**Weaknesses:**

*Main comments*

While the problem asked in the introduction on the difference between SGD and Adam (and its many variants) is an important problem where almost nothing is known, I find the answer of this article non-really convincing:
- The main phenomenon pinpointed by the authors is the presence of the constant $e^{\eta^2 \ell^2}$ in the bound to find a critical point: this factor is due to the fact that, initially, the step size is too big compared to the local curvature of the function while after some time, the step-size being a decreasing function $\eta_t = \eta/\sqrt{t}$, the step-size becomes well-conditioned. While some similar phenomenon may take place for some learning problems, I am not sure that such an analysis is the crux of the problem for the comparison between Adam and SGD. Maybe some experiments on non-toyish problem might help convince the reader (or at least myself): do we really see this $e^{\eta^2 \ell^2}$ popping out and eventually really slows down the convergence?
- The article claims to address, for practical purposes, and improve the theory from *bounded stochastic gradients* to *bounded variance* of the stochastic gradient: surely, theoretically it is a nice contribution, but it does not really serve to give an answer to the claimed question. Furthermore, even in this case, I think that the set-up is still not valid for the simplest least square case… so it does not seem to me as an incredible update.
- Finally, as stated in the strengths paragraph, the article addresses a lot of different setups, with different algorithms, sometimes stochastic, sometimes deterministic and it is very difficult to understand the true contribution of the article if the authors do not pinpoint them. Sometimes the reader, or at least I, was completely lost in what was known, both it terms of technique and/or result.

*Minor comments*

- Theorem 1: $\Delta$ is not defined. I am surprised that there is no problem when $\eta$ is too big (no upper bound on $\eta$, and it does not diverge!), but this may be an artifact on the bounded variance stochastic gradient assumption.
- Theorem 2: Good lower bound. I like it together with the illustration, as we understand well the phenomenon. However, why not proving it for SGD? This is only a technical limitation I guess.

*Final precaution.*

Overall, I have to say that I do not come from the community that analyses the general convergence of SGD for non-convex problems and its many normalized variants. Hence, it is hard to see what is the reel contribution of the authors.



**Questions:**

Already said in the paragraph above

**Limitations:**

Already said in the paragraph above

---

> ### Author Rebuttal · Authors · 2023-08-09
>
> Thanks for the comments!
>
> > **The main phenomenon ... Maybe some experiments on non-toyish problem ...**
>
> Thank you for suggesting additional experiments. We have now included a large-scale experiment on a widely-used 50-layer ResNet trained on the CIFAR-10 dataset and observed similar phenomenon.  With a larger initial stepsize, SGD tends to first experience an exponential blow-up before eventually converging as the stepsize decreases sufficiently. In contrast, adaptive gradient methods are robust to the stepsize change.
> Please find the experimental results in the uploaded PDF of the general response to all reviewers.
> We believe these experiments are representative and well support our theoretical findings on the benefits of adaptive methods over SGD.
>
> > **The article claims ... not really serve to give an answer to the claimed question ... the set-up is still not valid for the simplest least square case … it does not seem to me as an incredible update.**
>
> Bounded variance is a standard assumption in the literature (Ghadimi et al., 2013; Zaheer et al., 2018; Arjevani et al., 2023), which we believe is a good starting point to study the benefits of adaptive gradient methods over SGD. While this assumption may not cover all practical ML applications, we note that it holds for many important examples. For instance,  bounded variance assumption has been confirmed in certain deep learning tasks where stochastic gradient noises exhibit Gaussian-like patterns (Xie et al., 2020). In contrast, the bounded stochastic gradient is a much more restrictive assumption and does not hold even for Gaussian noises. From a theoretical standpoint, removing bounded gradient assumptions for adaptive methods is an important and active research topic (Faw et al., 2022; Kavis et al., 2022; Wang et al., 2023). Therefore, we believe that our efforts in moving beyond this assumption are a significant contribution.
> Furthermore, we expect our results can be extended to more relaxed assumptions, such as the expected smoothness (Khaled et al., 2022), defined as $\mathbb{E} \\|g(x, \xi)\\| \leq 2A (f(x) - f^\*) + B\\| \nabla f(x) \\|^2 + C$ for some constants $A, B, C \geq 0$, which accommodates the least square problem suggested by the reviewer.
>
>
> > **Finally ... it is very difficult to understand the true contribution of the article ... I, was completely lost in what was known ...**
>
>
> We investigate different algorithms under different setups in order to provide a comprehensive and sound comparison between untuned SGD and adaptive methods. Collectively, these results reveal that a strict advantage of adaptive methods is removing the exponential factor, not achieving an order-optimal rate parameter-agnostically as commonly perceived  -- an insight that we formally prove in this paper.
>
> As illustrated in Table 1, all entries in cells without accompanying theorem references are new results.
>
>
> We would like to highlight that the following are the most pivotal contributions of our paper:
>
> - For SGD, when using diminishing stepsizes, it can converge without knowing $\ell$. This convergence, however, comes with the cost of an exponential dependency on $\eta \ell$. We derive corresponding upper (Theorem 1 and 6) and lower (Theorem 2) bounds.
> - For AMSGrad-Norm, we provide convergence analysis for the deterministic setting with both upper and lower bounds (Theorem 5 and 8).
> In the stochastic setting, we show that its convergence can be arbitrarily slower than any polynomial rate (Theorem 4).
> - For Normalized SGD, we establish its nonconvergence in the stochastic settings with any stepsize (Theorem 3).
>
> The rest of our results are also novel and help support our key insight and build a complete picture. Following the reviewer's comment, we will refine the paper's structure and enhance its clarity in the new version.
>
>
> > **Theorem 1: $\Delta$ is not defined ... there is no problem when $\eta$ is too big ... this may be an artifact on the bounded variance stochastic gradient assumption.**
>
>
> The symbol $\Delta$ is defined in line 148 as the initial function value gap.
> The convergence with a large $\eta$ is not due to the assumption of bounded variance but rather the use of diminishing stepsizes.
> When $\eta$ is excessively large, SGD will initially diverge and then converge once the value of $t$ becomes large enough to render $\eta/\sqrt{t}$ small.
> The initial divergence behavior is encapsulated by the exponential dependence of $\eta\ell$ in the upper bound, i.e., $(4e)^{2 \eta^2\ell^2}$, which is the cost of excessively large initial stepsize.
>
>
> > **Theorem 2: Good lower bound ... However, why not proving it for SGD? ...**
>
> Thank you for recognizing the value of our lower bound.
> We note that the lower bound for GD is a stronger result. As GD is a special case of SGD when the gradient noise is reduced to zero, the lower bound in Theorem 2 also works for SGD. The lower bound already matches our upper bound in Theorem 1, implying that the lower bound is also tight for SGD.
>
>
> **References**
>
> - Ghadimi, Saeed, et al. "Stochastic first-and zeroth-order methods for nonconvex stochastic programming." SIAM Journal on Optimization. 2013.
> - Arjevani, Yossi, et al. "Lower bounds for non-convex stochastic optimization." Mathematical Programming. 2023.
> - Zaheer, Manzil, et al. "Adaptive methods for nonconvex optimization."  NeurIPS. 2018.
> - Khaled, Ahmed, et al. "Better Theory for SGD in the Nonconvex World." TMLR. 2022.
> - Xie, Zeke, et al. "A Diffusion Theory For Deep Learning Dynamics: Stochastic Gradient Descent Exponentially Favors Flat Minima." ICML. 2020.
> - Faw, Matthew, et al. "The power of adaptivity in sgd: Self-tuning step sizes with unbounded gradients and affine variance." COLT. 2022.
> - Kavis, Ali, et al. "Adaptive stochastic variance reduction for non-convex finite-sum minimization." NeurIPS. 2022.
> - Wang, Bohan, et al. "Convergence of AdaGrad for Non-convex Objectives: Simple Proofs and Relaxed Assumptions." COLT. 2023.

---

> > ### Comment · Reviewer_CaFA · 2023-08-14
> >
> > I thank the authors for the rebuttal and for the additional experiments. The authors have discussed my concerns but not really addressed my main ones: notably the significance of their result to help understand the behavior of gradient-adapted step-sizes.
> > While the additional experiments show the blow-up phase due to the large step-sizes at first, I wonder whether this type of learning curve really happen in practice: this is the first time I see personally an initial blow up.
> > For these reasons, I decide to maintain my score.

---

> > > ### Author Response · Authors · 2023-08-15
> > >
> > > We would like to emphasize the significance of our findings regarding the behavior of SGD versus adaptive methods and provide comments on the initial blowup.
> > >
> > > **Significance of Our Results:** Adaptive methods are frequently observed to converge fast and adeptly avoid large gradients, compared to SGD. Yet, there is a notable gap in the theoretical foundation supporting these observations, as we discussed in the "SGD vs. adaptive methods" section of our "related work." Our study stands out as one of the first to provide a quantitative analysis of this phenomenon, differentiating the sample complexities of untuned SGD from adaptive methods. Moreover, it aligns with several existing intuitions in the literature [23, 54, 26] that using gradient norms to rescale the stepsize can mitigate gradient explosion.
> > >
> > > **Initial Blowup:**
> > > The pattern of an initial blowup followed by convergence is less often highlighted in published works for a couple of reasons: (1) Most papers present results using the best-tuned stepsize. Typically, if practitioners notice an increasing gradient initially, they switch to a smaller stepsize. (2) The magnitude of the gradient explosion can sometimes surpass numerical limits, preventing the observation of subsequent convergence. This phenomenon is consistent with the exponential term in our theory and is also evident in figure (d) of our additional experiment. However, this behavior is not absent from the literature. For instance, it is documented in the following works:
> > >
> > > - The first and third subplots of Figure 2 in [Agarwal et al., 2022], in which BERT was trained using SGD and Adam.
> > > - Figure 2 of [Moulines et al., 2011], where they used a qudratic toy examples the same as the one utilized for the left subplot of our Figure 1.
> > >
> > >
> > > **References**
> > >
> > > - Agarwal, Naman, et al. "Learning Rate Grafting: Transferability of Optimizer Tuning." 2022 (https://openreview.net/forum?id=FpKgG31Z_i9).
> > >
> > > - Moulines, Eric, and Francis Bach. "Non-asymptotic analysis of stochastic approximation algorithms for machine learning." NeurIPS 2011.

---

### Author Rebuttal · Authors · 2023-08-09

We thank the reviewers for their valuable feedback and the overall positive evaluation of our work.

 As requested by Reviewers CaFA and vyoV, we conducted additional experiments with deep neural networks to demonstrate the gradient explosion effect on a more practical (large scale) experiment. In particular, we evaluated the performance of untuned SGD and several adaptive variants on the CIFAR-10 (Krizhevsky et al., 2009) dataset using a 50-layer ResNet (He et al., 2016). The observations are consistent: with a larger initial stepsize, SGD tends to first experience an exponential blow-up before eventually converging as the stepsize decreases sufficiently, while adaptive gradient methods are robust to the stepsize change. We kindly refer to the attached PDF below for more details.

If the reviewers have further questions, we will be happy to address them in the discussion phase.

**References**

- Krizhevsky, Alex, et al. "Learning multiple layers of features from tiny images." 2009.
- He, Kaiming, et al. "Deep residual learning for image recognition." CVPR. 2016.

---

### Decision · Program_Chairs · 2023-09-21

**Decision:**

Accept (poster)

**Comment:**

The paper presents valuable theoretical insights into the behavior of stochastic gradient descent (SGD) in the non-convex setting, specifically addressing, via lower bounds, potential exponential dependence on the smoothness constant when using untuned SGD. In contrast, the authors demonstrate that this exponential dependence can be avoided through adaptiveness in the optimization algorithm. While some concerns were raised regarding the empirical evaluation and clarity of contributions in relation to prior work, the authors have effectively addressed these concerns in their rebuttal. The theoretical contributions and the successful clarification of differences with existing literature warrant acceptance of the paper, which offers compelling insights into optimization algorithms' behavior and the value of adaptiveness in modern settings.